# Cell-specific and shared regulatory elements control a multigene locus active in mammary and salivary glands

Hye Kyung Lee [1] ✉, Michaela Willi [1], Chengyu Liu[2] & Lothar Hennighausen [1] ✉

Regulation of high-density loci harboring genes with different cell-specificities remains a puzzle. Here we investigate a locus that evolved through gene duplication and contains eight genes and 20 candidate regulatory elements, including one super-enhancer. *Casein* genes (*Csn1s1*, *Csn2*, *Csn1s2a*, *Csn1s2b*, *Csn3*) are expressed in mammary glands, induced 10,000-fold during pregnancy and account for 50% of mRNAs during lactation, *Prr27* and *Fdcsp* are salivary-specific and *Odam* has dual specificity. We probed the function of 12 candidate regulatory elements, individually and in combination, in the mouse genome. The super-enhancer is essential for the expression of *Csn3*, *Csn1s2b*, *Odam* and *Fdcsp* but largely dispensable for *Csn1s1*, *Csn2* and *Csn1s2a*. *Csn3* activation also requires its own local enhancer. Synergism between local enhancers and cytokine-responsive promoter elements facilitates activation of *Csn2* during pregnancy. Our work identifies the regulatory complexity of a multigene locus with an ancestral super-enhancer active in mammary and salivary tissue and local enhancers and promoter elements unique to mammary tissue.

Mammals arose from the synapsid lineage about 160 million years ago in the middle Jurassic, and all extant mammals rely entirely on milk to nourish their young[1]. The transformation of proto-lacteal fluid rich in calcium-binding proteins into nutritious milk was vital for the sustained success of mammals. The innovation of milk and saliva[2,3] was largely driven by the expansion of the secretory calcium-binding phosphoprotein (SCPP) gene family, the acquisition and evolution of genetic regulatory elements conveying gene expression to either mammary or salivary secreting cells and the diversification and repurposing of proteins. The five caseins (CSN1S1, CSN2, CSN1S2A, CSN1S2B, CSN3) are cardinal milk proteins accounting for up to 80% of total milk protein[4], and their gene arrangement within the casein locus partially reflects its evolutionary history with *Csn3* juxtaposed to its ancestral precursor *Fdcsp* (Follicular dendritic cell secreted protein) and *Odam* (Odontogenic ameloblast associated) [5,6].

The extended casein locus is of particular interest as it harbors at least eight genes expressed in mammary or salivary glands or both[3,5,6]. While the five *Casein* genes are dominantly expressed in mammary tissue and highly activated during pregnancy, *Fdcsp* is preferentially active in salivary tissue and in immune cells, and *Odam* is expressed in mammary and salivary tissues. The casein locus provides a unique window into the evolution of a multigene locus and co-evolving regulatory elements targeting the expression of individual genes to at least four distinct cell types, mammary, salivary, tooth-associate tissues and specific B cells.

Over the past 25 years, major mammary-centric hormones, receptors, tyrosine kinases and transcription factors have been deleted in mice resulting in dramatic defects in mammary development and function[7–13], frequently leading to lactation failure because of the absence of a differentiated secretory alveolar compartment. However,

[1]Section of Genetics and Physiology, Laboratory of Cellular and Molecular Biology, National Institute of Diabetes and Digestive and Kidney Diseases, US National Institutes of Health, Bethesda, Maryland 20892, USA. [2]Transgenic Core, National Heart, Lung, and Blood Institute, US National Institutes of Health, Bethesda, Maryland 20892, USA. ✉e-mail: hyekyung.lee@nih.gov; lotharh@niddk.nih.gov

a major challenge is discerning primary effects from secondary ones. For example, loss of the transcription factor STAT5 prevents the activation of alveolar differentiation programs during pregnancy and the failure to launch milk protein gene expression might be, in some cases, a secondary consequence. Thus, understanding the control mechanisms of genes during normal physiology requires the inactivation of candidate regulatory elements rather than the global or cell-specific ablation of transcription factors.

Our data presented here on the individual and combinatorial deletion of 12 regulatory elements within a locus encoding eight genes provides evidence of a complex super-enhancer active in mammary and salivary tissue, gene-specific local enhancers and a cytokine-responsive promoter that activate genes 10,000-fold in mammary tissue during pregnancy.

## Results

### Enhancer structure in a locus active in mammary tissue

The extended casein locus spanning approximately 330 kbp encodes at least eight genes (Fig. 1a) with distinct expression patterns. While the five *Casein* genes (*Csn1s1, Csn2, Csn1s2a, Csn1s2b,* and *Csn3*) are known to be expressed almost exclusively in mammary tissue, *Fdcsp* and *Odam* are active preferentially in salivary glands. First, we gauged the absolute and relative RNA levels of these genes in mammary glands at day one of lactation (L1) and salivary tissue through RNA-seq (Fig. 1b). Combined, the mRNA levels of the five *caseins* account for more than 50% of mRNA in mammary tissue, and up to $10^6$ reads were recorded for individual *Casein* genes. *Casein* gene expression in salivary tissue is four to five orders of magnitude lower. While expression of *Fdcsp* is confined to salivary tissue, *Odam* mRNA levels are similar in both mammary and salivary tissue. Multispecies comparisons (Supplementary Fig. 1a) highlighted the complexity of this locus[5] and identified an evolutionary conserved region as a candidate regulatory element activating this locus during pregnancy[14,15].

Next, we measured the expression of these genes throughout mammary development, from non-parous mice to day 10 of lactation (L10) (Fig. 1c). mRNA levels for the five caseins ranged from 100 to 1000 normalized read counts in virgin mice and peaked at approximately $10^7$ normalized read counts at L10, an increase of four to five orders of magnitude. Overall, the most prominent induction was between day six of pregnancy (p6) and L1, the time window of heightened placental lactogen- and prolactin-mediated epithelial cell differentiation. Another 10-fold induction was observed between L1 and L10 (Supplementary Data 1). In contrast, *Odam* expression was only observed at L1 and L10, and mRNA levels were four to five orders of magnitude lower than the caseins.

The structure of this locus, with salivary genes nestled between mammary genes, makes it an ideal case study on cell-specific and hormone-activated regulatory elements. Candidate regulatory elements were identified based on the presence of H3K27ac marks and transcription factor (TF) occupancy (Fig. 1d, e). As anticipated, H3K27ac marks at the five *Casein* genes were restricted to mammary tissue. However, H3K27ac coverage upstream of *Odam*, the ancestral gene of the locus, was detected in both mammary and salivary tissue, pointing to shared regulatory elements operative in both tissues. This region also harbors a 147 bp long evolutionarily conserved region (ECR) identified in mammals[5].

Digging deeper, we explored binding of mammary-centric TFs[16], such as the cytokine-inducible transcription factor Signal Transducer and Activator of Transcription (STAT) 5, the glucocorticoid receptor (GR), Nuclear Factor IB (NFIB) and mediator complex subunit 1 (MED1) (Fig. 1e). TF binding coincided with H3K27ac and H3K4me1 marks and a total of 20 candidate regulatory elements were identified within the 330 kb locus. In addition, CTCF binds to four distinct sites within this locus[17]. The H3K27ac marked region upstream of *Odam* contains four STAT5-bound regions and has all hallmarks of a super-enhancer (SE) as

defined by the Rose algorithm (Fig. 1e, area highlighted in red). RNA polymerase II (Pol II) coverage was most prominent in the *Casein* genes. Integration of active histone marks, transcription factor binding, and gene expression data suggest that the mammary-salivary locus harbors the highest density of candidate enhancers and highly regulated genes among all multigene loci in mammary tissue (Supplementary Fig. 1b–d, Supplementary Table 1).

Although all 20 candidate regulatory elements were bound by STAT5, a principal TF controlling mammary development and function[8], only 12 STAT5 peaks coincided with genuine DNA binding motifs (GAS, interferon-Gamma Activated Sequence) (Supplementary Table 2), suggesting that STAT5 binding at other sites is mediated by other TFs, such as NFIB or GR. An inherent diversity in anchor proteins could lead to seemingly identical enhancers, with possibly different and unique activities. STAT5 binding was also detected at the promoter regions of *Csn1s1, Csn2, Csns2a* and *Csn1s2b* and coincided with bona fide H3K4me1 enhancer marks[18,19] (Supplementary Fig. 2). As expected, H3K4me3 marks were exclusively associated with promoter regions and Pol II coverage was preferentially over gene bodies.

### Super-enhancer activity in mammary tissue during pregnancy

To identify dual regulatory elements controlling genes in mammary and salivary glands, we focused on the candidate SE located between *Csn1s2b* and *Odam,* the ancestral gene expressed in both tissues. The SE is composed of four constituent enhancer modules (SE-E1, SE-E2, SE-E3 and SE-E4) spanning a total of 10 kbp (Fig. 2a and Supplementary Fig. 3) and we generated mice carrying individual and combinatorial enhancer deletions (Supplementary Fig. 4). Deletion of the entire 10 kbp SE (ΔSE) was confirmed by sequencing and the absence of TF binding and H3K27ac marks in this region (Fig. 2c). Female mice lacking the SE were unable to support their litters due to lactation failure and analyses were conducted in mammary tissue at day 18 of pregnancy (p18), just prior to delivery or within 12 h after delivery (post-partum <12 h, pp <12 h). This was critical as lactation failure results in the loss of cell differentiation and loss of STAT5 phosphorylation and tissue remodeling is observed approximately 24 h after parturition in a mouse model[20]. Of note, the C57BL/6 strain used in this study has a gestation period of 18.5 days[21]. Although we had hypothesized that the entire shared casein locus would be under SE control, distinct gene-specific differences were observed. *Odam* mRNA levels were reduced by more than 99%, *Csn3* by 98% and *Csn1s2b* by 93% (Fig. 2b and Supplementary Data 2), which coincided with a decline of H3K27ac, H3K4me3, and Pol II coverage at these genes, but not in *Wap,* another milk protein gene (Fig. 2c and Supplementary Fig. 3). *Csn1s1, Csn2* and *Csn1s2a* expression was preserved at approximately 50%, 70% and 60% of wild-type (WT) levels, respectively. Since the sequencing depth of ChIP-seq libraries can influence the peak heights of TF binding and of histone modifications, we also present ChIP-seq images of the STAT5 target gene *Cish* located on chromosome 9 and that are therefore not influenced by the SE located on chromosome 5 (Fig. 2c, right panel) (see Material and Methods for details). In contrast, *Csn2* and *Csn1s2a* mRNA levels were reduced at a lower, yet statistically significant, level. This experiment demonstrates that the SE dictates the expression of three genes but has a limited impact on the other three mammary genes in the shared locus, whose regulation might be controlled by gene-specific enhancers.

To understand whether the SE is required for the establishment of gene-specific regulatory elements, such as enhancers, we conducted ChIP-seq experiments in tissue lacking the SE. STAT5, GR and NFIB binding at the two candidate *Csn3* enhancers remained intact in the absence of the SE (Fig. 2c), suggesting the inability of local enhancers to functionally compensate for the SE. In contrast, STAT5 binding at the *Csn1s2b* proximal enhancer is lost in ΔSE mammary tissue suggesting that the SE activates this secondary gene-specific enhancer.

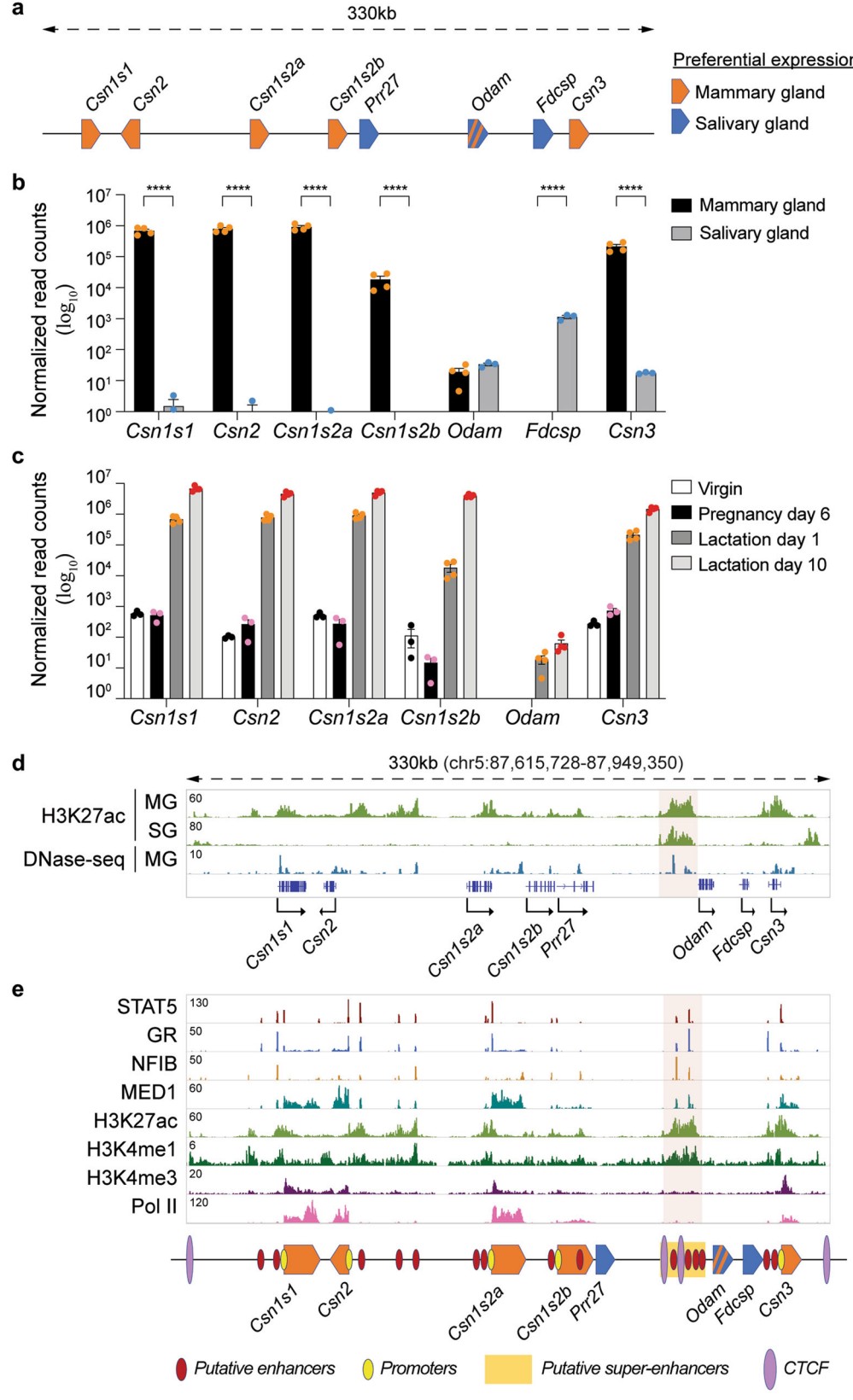

A defining feature of milk protein genes is their exceptional response to pregnancy hormones, first and foremost prolactin. While the SE differentially affects *Casein* genes after a full pregnancy, it might have a more extended function in early pregnancy, prior to the prolactin surges that activate milk protein genes. Expression data obtained from mammary tissue on day six of pregnancy (p6)

indicate an expanded SE function extending throughout the entire shared locus. In the absence of the SE (ΔSE), expression of *Odam* and all five *Casein* genes was reduced by more than 96% (Fig. 2d, e and Supplementary Data 3), suggesting a possible role in the establishment of activating chromatin marks and possibly TF binding in early pregnancy. While H3K27ac marks were present at some *Casein*

**Fig. 1 | Characteristics of the _Casein_ locus. a** Diagram presents gene structure within the mouse _Casein_ locus and their preferential expression. **b** mRNA levels of genes in the _Casein_ locus were measured by RNA-seq in lactating mammary glands (lactation day one, L1) and salivary glands ($n = 4$ and 3, respectively). Results are shown as the means ± SEM of independent biological replicates. _p_-values are from 2-way ANOVA with Sidak's multiple comparisons test. $**p < 0.01$, $****p < 0.0001$. **c** mRNA levels of _Csn_ genes were measured by RNA-seq at virgin, day six of pregnancy (p6), lactation day one (L1) and L10 (virgin, p6, $n = 3$; L1, L10, $n = 4$). Results are shown as the means ± SEM of independent biological replicates. _p_-values are

from 2-way ANOVA with Tukey's multiple comparisons test. **d** Chromatin features of the _Casein_ locus were identified by ChIP-seq and DNase-seq data in lactating mammary glands (lactation day one, L1) and salivary glands. The red shade indicates the super-enhancer. MG mammary gland, SG salivary gland. **e** ChIP-seq data for TFs binding and histone marks provided structural information of the _Casein_ locus at day one of lactation (L1). Red, yellow and purple circles mark candidate enhancers, promoters, and CTCF binding sites, respectively. Super-enhancer is indicated by yellow, a rectangle and a red shade. Source data are provided as a Source Data file.

genes at p6, they were greatly reduced in the absence of the SE (Fig. 2e).

_Casein_ mRNA levels in mammary tissue from non-parous female mice are equivalent to those obtained in the absence of the SE at p6, suggesting that the SE is biologically inactive at early pregnancy. An earlier study demonstrated enrichment of H3K4me2[22] over the region we now call SE-E1, and our own data demonstrate NFIB binding to SE-E1 in virgin tissue (Supplementary Fig. 5). However, we did not detect any H3K27ac marks, suggesting that the SE has no or only marginal activity at that stage.

The 10 kbp SE region harbors four areas bound by mammary TFs, and their capacity to function individually or in combination and contribute to the overall SE activity was not clear. To gain insight into the complexity of this SE, we introduced individual and combined deletions (Supplementary Fig. 4). Deletion of SE-E1 (ΔSE-E1), SE-E2 (ΔE2) or SE-E4 (ΔE4) resulted in the loss of STAT5 binding and H3K27ac at their respective sites. Notably, the establishment of SE-E3 and SE-E4 is dependent on the presence of SE-E2. ChIP-seq data from the STAT5 target gene _Cish_ were added to demonstrate the sequencing depth of the libraries (Supplementary Fig. 4A, right panel). While the loss of SE-E4 (ΔE4) had no discernible consequence on any of the _Casein_ genes, _Csn1s1_, _Csn1s2b_ and _Csn3_ mRNA levels were reduced between 40–70% in both ΔE1 and ΔE2 tissues. Combined deletion of both SE-E1 and SE-E2 (ΔE1/2) silenced the entire SE and mimicked the ΔSE mutation suggesting redundancy between SE-E1 and SE-E2. Of note, ΔE1/2 mice exhibited lactation failure.

Having identified the physiological significance of the SE in mammary tissue, we addressed its regulatory significance in salivary glands. While the loss of SE activity led to a complete silencing of _Odam_ expression, _Fdcsp_ and _Csn3_ mRNA levels declined by 99% and 88%, respectively (Fig. 3 and Supplementary Data 4), demonstrating its dual specificity.

## The _Fdcsp_ gene fails to be activated in mammary tissue

The follicular dendritic cell secreted protein (_Fdcsp_) gene is expressed in salivary glands and immune cells (https://www.ncbi.nlm.nih.gov/gene/260436) but not in mammary glands. The absence of _Fdcsp_ expression in mammary tissue might be the result of _Odam_ blocking the SE from efficiently reaching and activating the _Fdcsp_ promoter. To test this hypothesis, we deleted the _Odam_ gene, thus transporting the SE within a few kbp to the _Fdcsp_ gene (Fig. 4 and Supplementary Data 5). Despite being in the physical orbit of the SE, the _Fdcsp_ promoter remained silent in lactating mammary tissue (Fig. 4a, c), and expression in salivary tissue was unaltered (Fig. 4b, d). These findings suggest that the promoter is unresponsive to the SE. In contrast, _Csn3_ mRNA levels increased approximately 2-fold in mammary tissue, suggesting a distance-dependency of SE activity. No expression changes were observed in salivary glands.

To understand the mechanism behind _Fdcsp_ being nonreceptive to the powerful casein SE, we conducted global DNA methylation analyses in mammary tissue at day one of lactation and in liver tissue (Supplementary Fig. 6). The methylation status of the promoter regions (400 bp surrounding the transcriptional start site, TSS) correlated with their activities in mammary tissue. While the highly active _Casein_ genes displayed a low methylation coverage, the _Fdcsp_

promoter was fully methylated, reflecting a repressive state. The methylation status of specific mammary cell populations from Balb/C mice has been investigated[23], and we used these data to confirm the methylation status of the casein locus (Supplementary Fig. 6). In contrast to mammary tissue, the _Casein_ gene promoter regions were highly methylated in liver tissue.

## Local enhancers control _Csn3_ expression

Removal of the SE ablates _Csn3_ expression without impacting TF binding at the two _Csn3_ enhancers, thus questioning their physiological roles. We addressed this issue and introduced deletions within the distal (Csn3-E1) and proximal (Csn3-E2) _Csn3_ candidate enhancers (Fig. 5a). Strong STAT5 binding coinciding with two GAS motifs and an NFIB site occurred at Csn3-E2, and weaker binding was observed at Csn3-E1. GR binding was stronger at Csn3-E1 compared to Csn3-E2, suggesting distinct molecular structures and possibly functions of the two enhancers. While deletion of the distal candidate enhancer (ΔE1) resulted in the loss of TF binding and H3K27ac coverage at this site, _Csn3_ gene expression remained at 55% (Fig. 5b, c). Ablation of the Csn3-E2 enhancer resulted in lactation failure similar to CSN3 deficient mice[24], and gene expression was measured in mammary tissue at day 18 of pregnancy (p18), just prior to delivery or within 12 h after delivery (post-partum <12 h, pp <12 h), as in the Csn-SE mutant mice. Deletion of the two GAS motifs in Csn3-E2 (ΔE2-S) reduced _Csn3_ mRNA levels by 98%, revealing the reason for lactation failure. While STAT5 binding was completely abolished, residual binding of NFIB was detected in mammary tissue within 12 h after delivery (pp <12 h). Deletion of the two GAS motifs and the NFIB site (ΔE2-S/N) reduced _Csn3_ mRNA levels even further, coinciding with a complete absence of STAT5, GR and NFIB binding, loss of H3K27ac marks and Pol II loading (Fig. 5b, c). ChIP-seq data from the _Cish_ locus are shown as a reference. Of note, the absence of Csn3-E2 did not adversely affect other _Casein_ genes (Supplementary Fig. 7 and Supplementary Data 6). Interactions between the _Csn3_ enhancers and the SE had been confirmed by 3C, thus supporting their crosstalk[25].

## Local enhancer modifies _Csn1s1_ expression

Expression of _Csn1s1_, positioned at the 5′ border of the casein locus, is only marginally influenced by the SE located 200 kbp 3′ of the gene (Fig. 1), suggesting the presence of independent regulatory elements. H3K27ac marks pointed to candidate regulatory elements at −11.5 kb (Csn1s1-E1) and −3.5 kb (Csn1s1-E2) (Fig. 6a). Strong STAT5, GR and NFIB occupancy was detected at site Csn1s1-E2 but less so at Csn1s1-E1, which also had reduced H3K27ac coverage. STAT5 binding was also observed at the _Csn1s1_ promoter and coincided with a GAS motif at position −100 bp. While deletion of the GR motif in Csn1s1-E1 (ΔE1) resulted in the loss of GR and STAT5 binding at this site and reduced STAT5 binding at the promoter site, _Csn1s1_ expression at day one of lactation was not impaired (Fig. 6b, c). In contrast, deletion of the NFIB sites in Csn1s1-E2 (ΔE2) resulted in a 65% reduction of _Csn1s1_ expression and coincided with the loss of TF binding and H3K27ac at both enhancers. Despite the reduction of _Csn1s1_, lactation in these mice was overtly normal. These findings demonstrate that individually the two _Csn1s1_ enhancers have a very limited biological activity compared to the _Csn3_ enhancer described in this study. It is possible that the two

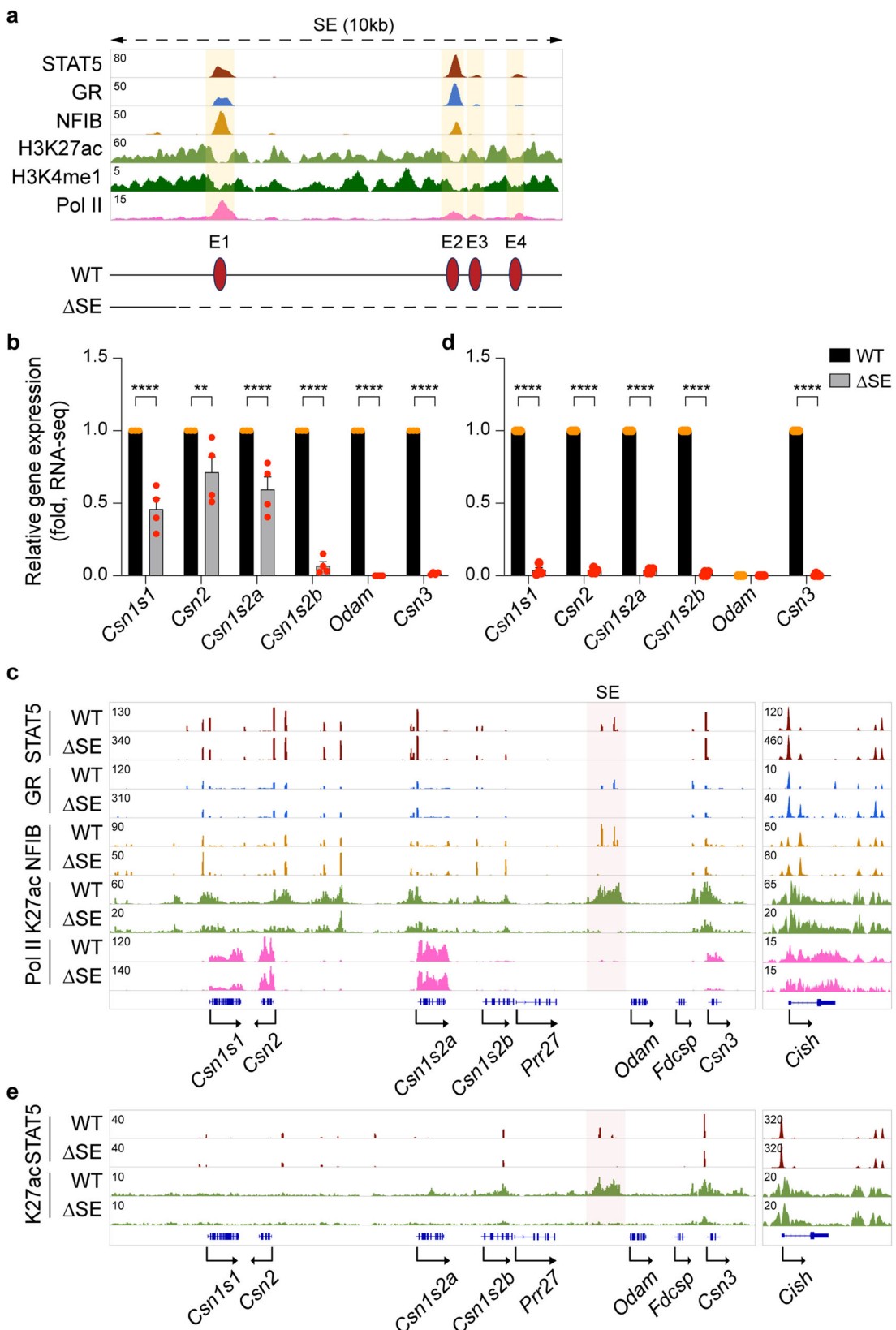

enhancers could be partially redundant and work in a superadditive fashion so that only when both enhancers are removed, the effect on *Csn1s1* expression would be revealed (see Discussion). Alternatively, the two candidate enhancers could synergize with the STAT5-based promoter element and control *Csn1s1* expression during pregnancy and lactation.

## Limited biological activity of the *Csn2* local enhancers

Although major investments have been made in understanding enhancer function, knowledge of promoter elements and their possible interactions with enhancers is limited. To gain insight into the possible contribution of promoter elements in the regulation of mammary genes, we investigated the regulation of the *Csn2* and

**Fig. 2 | Differential activation of selected *Casein* genes by the super-enhancer during pregnancy. a** The candidate super-enhancer was identified by ChIP-seq for TFs and activating histone marks in mammary tissue of wild-type (WT) mice at day one of lactation (L1). The yellow shades and red circles indicate candidate enhancers. The diagram shows the enhancer deletions introduced in mice using CRISPR/Cas9 genome engineering. **b** Expression of *Casein* genes was measured in mammary tissue of WT and ΔSE mice collected at day 18 of pregnancy (p18) by RNA-seq ($n = 4$). Results are shown as the means ± SEM of independent biological replicates. *p*-values are from 2-way ANOVA followed by Sidak's multiple comparisons test between WT and mutants. ***p* < 0.01, *****p* < 0.0001. **c** STAT5 and GR binding, and

H3K27ac and the Pol II coverage of the *Casein* locus in WT and ΔSE mice were monitored by ChIP-seq. Mammary tissues of WT and ΔSE mice were collected within 12 h post-partum (pp <12 h). The red shade indicates the super-enhancer (SE). The *Cish* locus served as ChIP-seq control. **d** Expression of *Casein* genes in mammary tissue of WT and ΔSE mice at day 6 of pregnancy (p6) was analyzed by RNA-seq (WT, $n = 3$; ΔSE, $n = 4$). Results are shown as the means ± SEM of independent biological replicates. *p*-values are from 2-way ANOVA followed by Sidak's multiple comparisons test between WT and mutants. *****p* < 0.0001. **e** Genomic characteristics of the *Casein* locus were determined by ChIP-seq for STAT5 and H3K27ac in WT and ΔSE tissues at p6. Source data are provided as a Source Data file.

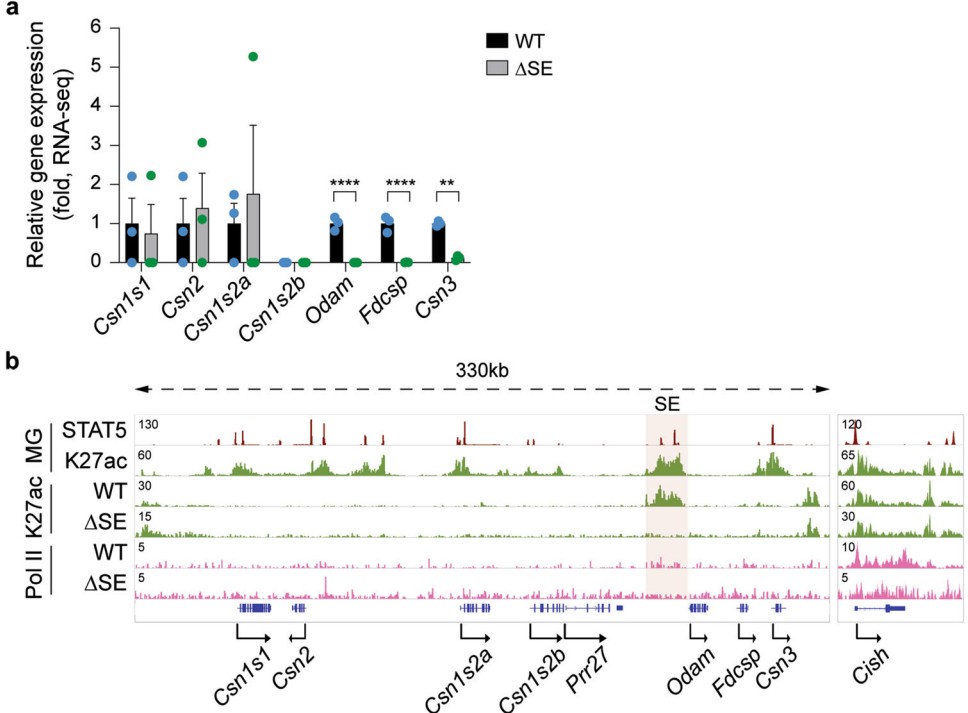

**Fig. 3 | Salivary-specific activation of selected genes in the *Casein* locus by the super-enhancer. a** Expression of the five *Casein* genes, *Odam* and *Fdcsp* genes was measured by RNA-seq in salivary tissue from ΔSE mice ($n = 3$). Results are shown as the means ± SEM of independent biological replicates. *p*-values are from 2-way ANOVA followed by Sidak's multiple comparisons test between WT and mutants.

***p* < 0.01, *****p* < 0.0001. **b** ChIP-seq analysis shows the H3K27ac and Pol II landscapes in salivary tissue from WT mice and mice lacking the super-enhancer (ΔSE). The red shade indicates the super-enhancer (SE). The red shade indicates the super-enhancer. The *Cish* locus served as ChIP-seq control. MG, mammary gland at day one of lactation (L1). Source data are provided as a Source Data file.

*Csn1s2a* genes that are transcribed in opposite directions and whose promoters are separated by 75 kbp (Fig. 7a). Both genes are activated approximately 10,000-fold between virgin and lactating mammary tissue and their combined mRNAs account for approximately 20% of total mRNA during lactation (Fig. 1c and Supplementary Data 1). The mammary transcription factors STAT5, GR and NFIB bind at five candidate enhancers (Csn2-E1 to Csn2-E5) that also coincide with activating histone marks (Fig. 7a). Canonical STAT5 recognition sites (GAS motifs) are present in only four enhancers, suggesting that binding to Csn2-E3 is mediated by another TF, possibly GR or NFIB (Supplementary Table 2). Of note, STAT5 binds to two distinct regions in Csn2-E1, Csn2-E2, and Csn2-E4, suggesting additional internal complexity. STAT5 and GR binding was also associated with the *Csn2* promoter, coinciding with a canonical ($TTCn_3GAA$) and a non-canonical ($TTCn_4GAA$) GAS motif, as well as with the *Csn1s2a* promoter. No bona fide GR binding site was detected, suggesting GR binding mediated by STAT5. The H3K27ac patterns differed between enhancer and promoter elements. While TF binding at enhancers was within an H3K27ac 'gap', binding at promoters was outside a classical histone-depleted gap. This points to the possibility that TF binding at

enhancers and promoters and their interactions with H3K27ac marks is distinctly different and might serve different purposes. It is further possible that promoter regions are more flexible and present higher heterogeneity in TF binding and different chromatin organization due to the gene transcription or 3D organization of these genes can be envisioned.

First, we investigated the physiological role of the three candidate enhancers located upstream of *Csn2*, Csn2-E1 at −6 kbp, Csn2-E2 at −25 kbp and Csn2-E3 at −35 kbp (Fig. 7a). All three candidate enhancers are bound by STAT5, GR and MED1. We used CRISPR/Cas9 genome engineering to delete these enhancers and monitor their functional significance in mammary tissue on day one of lactation (L1). Neither the deletion of Csn2-E1 nor the combined deletion of Csn2-E2 and Csn2-E3 resulted in a significant reduction of *Csn2* mRNA levels (Fig. 7b). The combined deletion of all three candidate enhancers resulted in an approximately 50% decrease in *Csn2* mRNA levels. The deletion of enhancers was validated by DNA sequencing and ChIP-seq analyses for activating histone marks, STAT5 and GR (Fig. 7c). Expression of *Csn1s2a* was not significantly impaired in these mutants, and TF binding to candidate elements associated with *Csn1s2a* was not

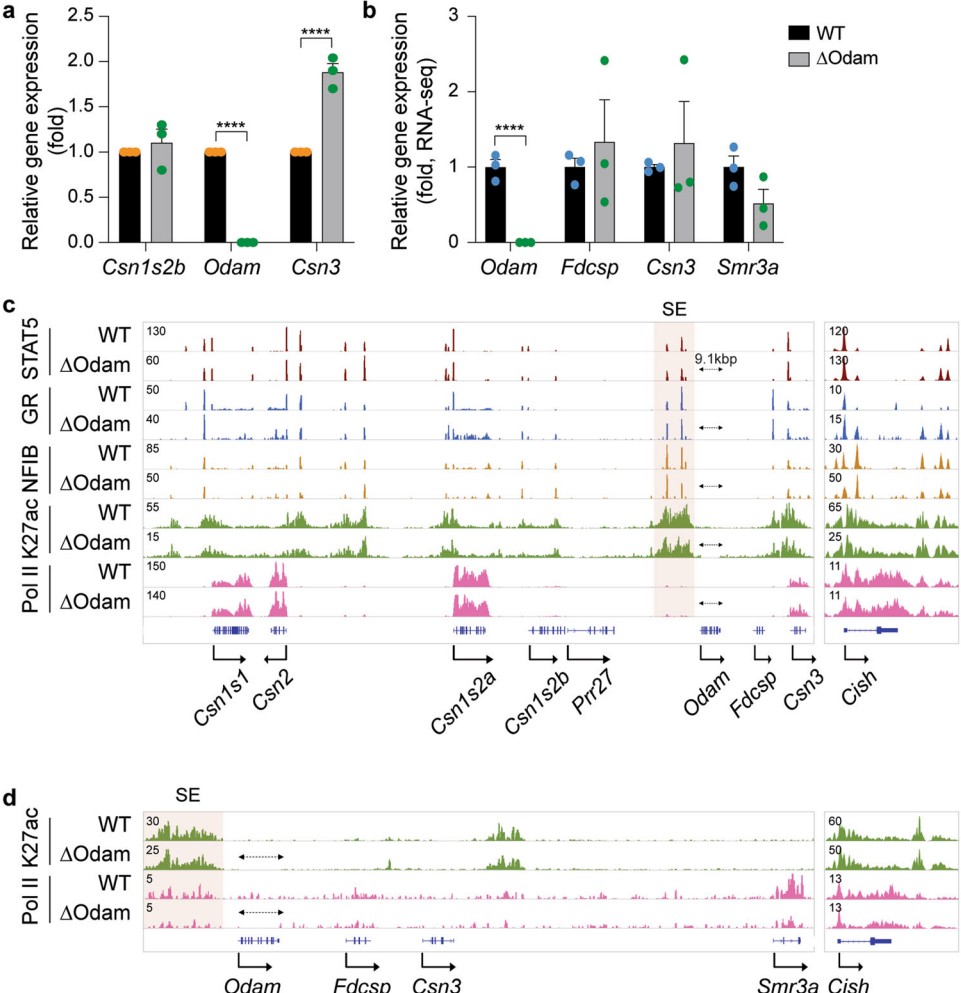

**Fig. 4 | Distance-dependent function of the super-enhancer. a** *Odam* and *Csn3* mRNA levels were measured by qRT-PCR in lactating mammary tissues (day one of lactation, L1), from WT mice and mutant mice after deletion of the Odam gene (ΔOdam) (normalized to *Gapdh* levels). Results are shown as the means ± SEM of independent biological replicates (*n* = 3). 2-way ANOVA followed by Sidak's multiple comparisons test was used to evaluate the statistical significance of differences in WT and mutants. ****$p < 0.0001$. **b** mRNA levels of salivary expressed gene in the *Casein* locus were measured by RNA-seq (*n* = 3). **c, d** Genomic features of the Odam-*Csn3* locus were characterized by ChIP-seq in lactating mammary (day one of lactation, L1) (**c**) and salivary (**d**) tissues from WT and ΔOdam mutant mice. The red shade indicates the super-enhancer (SE). The *Cish* locus served as ChIP-seq control. Source data are provided as a Source Data file.

affected. The *Cish* locus was used as a reference for the sequencing depth.

## Synergy between promoter and enhancer elements

Based on our results, the three distal enhancers contribute only modestly to the activation of the *Csn2* gene, opening the possibility of additional regulatory elements. We therefore focused on the two STAT5 binding sites within the *Csn2* promoter region (Fig. 8a) and introduced disabling point mutations into the palindromic sequences of the canonical and non-canonical GAS motifs located within 150 bp of the transcriptional start site. Disabling both GAS sites (ΔP) through deaminase base editing resulted in an approximately 80% reduction of *Csn2* mRNA at L1 (Fig. 8b and Supplementary Data 7). These findings begged the question of potential synergy between the cytokine-responsive promoter elements and the three distal enhancers. We addressed this in mice lacking the three enhancers and carrying point mutations in the two promoter-based GAS elements. The combined mutations resulted in an almost complete silencing of the *Csn2* gene, with expression levels reduced by four orders of magnitude (99.99%) (Fig. 8b). Loss of these elements coincided with the complete absence of Pol II coverage and H3K27ac marks (Fig. 8c). The presence of the three enhancers and the

promoter-based STAT5 sites is required for the full activation of Csn2 expression during lactation (Supplementary Fig. 8a and Supplementary Data 8), but their loss does not affect basal expression in mammary tissue of non-parous (virgin) mice (Supplementary Fig. 8b and Supplementary Data 9). This is further evidence that these regulatory elements elicit hormone responsiveness.

There is consensus that all seven STATs, except for STAT6, activate gene expression through canonical GAS motifs. A single STAT5 peak covers the closely spaced canonical and non-canonical GAS motifs in the *Csn2* promoter (Supplementary Fig. 9a), making it difficult to discern if either site or both sites are bound by STAT5 and convey transcriptional activity. To distinguish the relative significance of the two GAS motifs, we introduced mutations individually into the two sites, both in the presence and absence of the three distal enhancers (Supplementary Fig. 9a). In the presence of the distal enhancers, mutation of the canonical site (A) did not impact *Csn2* expression, and a 50% reduction was observed upon the simultaneous deletion of the three enhancers (Csn2ΔP-E1/2/3-A) (Supplementary Fig. 9b). Introduction of a disabling mutation into the non-canonical site (B) resulted in a ~80% reduction of *Csn2* mRNA levels and the additional absence of the distal enhancers (Csn2-ΔP-E1/2/3-B) resulted

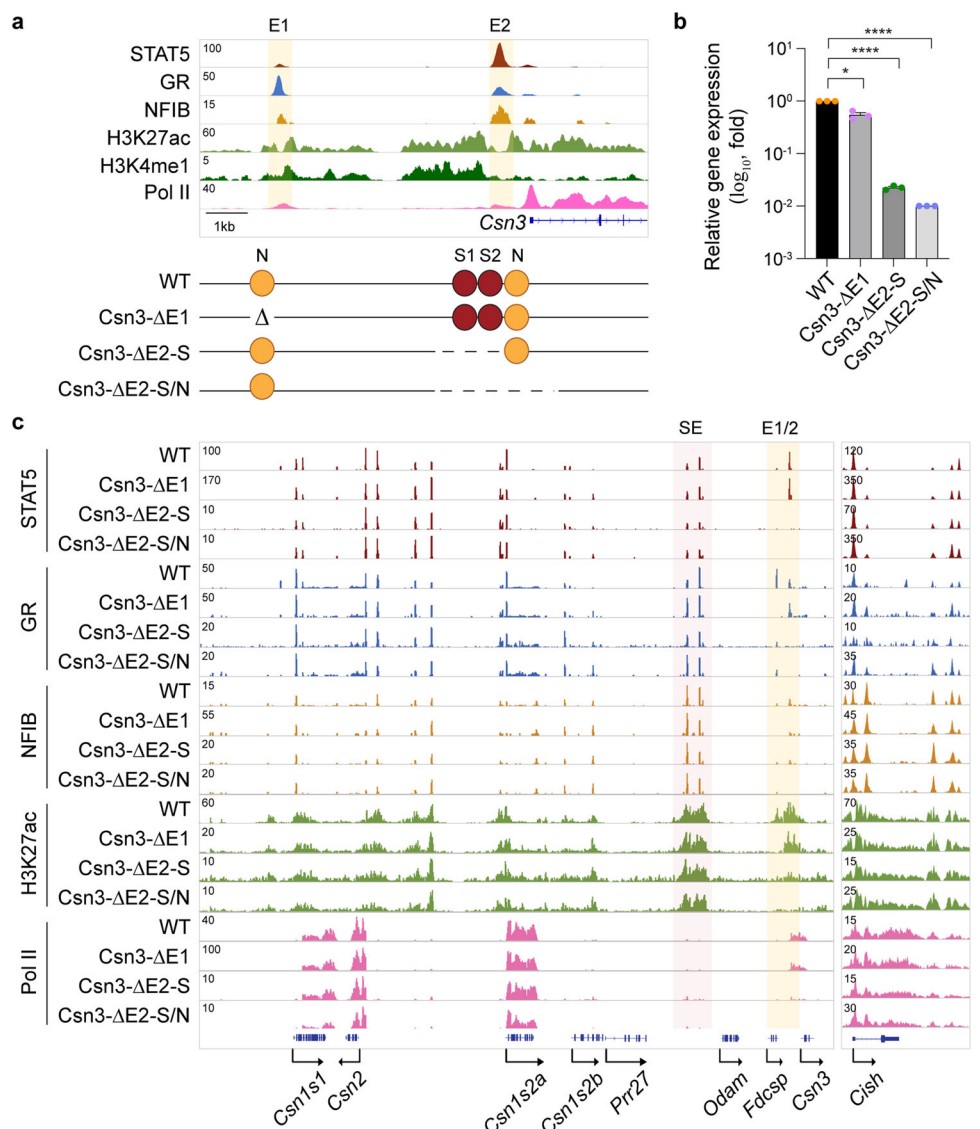

**Fig. 5 | Super-enhancer-dependent local enhancers activate *Csn3* expression.**
**a** The presence of H3K27ac and H3K4me1 marks in mammary glands of wild-type mice at day one of lactation (L1) indicates a distal candidate enhancer (Csn3-E1) at −7 kbp and a proximal one (Csn3-E2) at −0.6 kbp. The yellow shade indicates the enhancers. The diagram shows the enhancer deletions introduced in mice using CRISPR/Cas9 genome engineering. Red and orange circles indicate the GAS motif and NFIB binding sites. **b** *Csn3* mRNA levels were measured by qRT-PCR in pregnancy day 18 (p18) mammary tissue from WT mice and mice lacking the *Csn3* distal enhancer (ΔE1) and *Csn3* proximal enhancer (ΔE2) and normalized to *Gapdh* levels.

Results are shown as the means ± SEM of independent biological replicates ($n = 3$). Two-tailed $t$-test with Welch's correction was used to evaluate the statistical significance of differences between WT and each mutant mouse line. *$p < 0.05$, ****$p < 0.00001$. **c** Genomic features of the entire *Csn* locus were investigated by ChIP-seq in lactating mammary tissue (collected within 12 h post-partum (pp <12 h)) of WT, ΔE1, ΔE2-S and ΔE2-S/N mice. The red shade indicates the super-enhancer (SE). The *Cish* locus served as ChIP-seq control. Source data are provided as a Source Data file.

in the complete loss of *Csn2* expression and levels were four orders lower than in mice with intact regulatory elements. STAT5 binding was still observed upon loss of the canonical GAS motif and completely absent in the non-canonical mutant (Supplementary Fig. 9c). These results provide evidence that the non-canonical GAS motif is a key element in the *Csn2* promoter and STAT5 preferentially binds at the non-canonical site.

## Discussion

Although mouse genetics has been used to address the functionality of enhancers in single gene loci[16,26–29], key questions pertaining to the contribution of enhancers and super-enhancers (SE) to gene regulation remain to be answered. Specifically, understanding the regulation of complex multigene loci harboring genes expressed in one or more

distinct cell types is lacking. Our study provides insight into regulatory mechanisms operative in salivary and mammary gland tissues that share morphological and molecular features during embryogenesis[30–32]. Specifically, we identified a SE active in mammary and salivary tissues, local gene-specific enhancers and a cytokine-responsive promoter element that activate transcription during pregnancy. The shared locus, with its eight genes linked to lactation, saliva and immune response[6], is an evolutionary playground that fostered regulatory innovation with 20 enhancer and promoter elements. *Odam* and its associated SE likely constitute the ancestral unit of the shared locus and regulatory activity expanded from salivary tissue to mammary tissue. However, as this locus expanded, the five newly formed *Casein* genes acquired their own regulatory elements and three gained partial independence of the SE, at least during lactation (Fig. 9a, b). Although *Csn1s2b*[25] and *Csn3* acquired

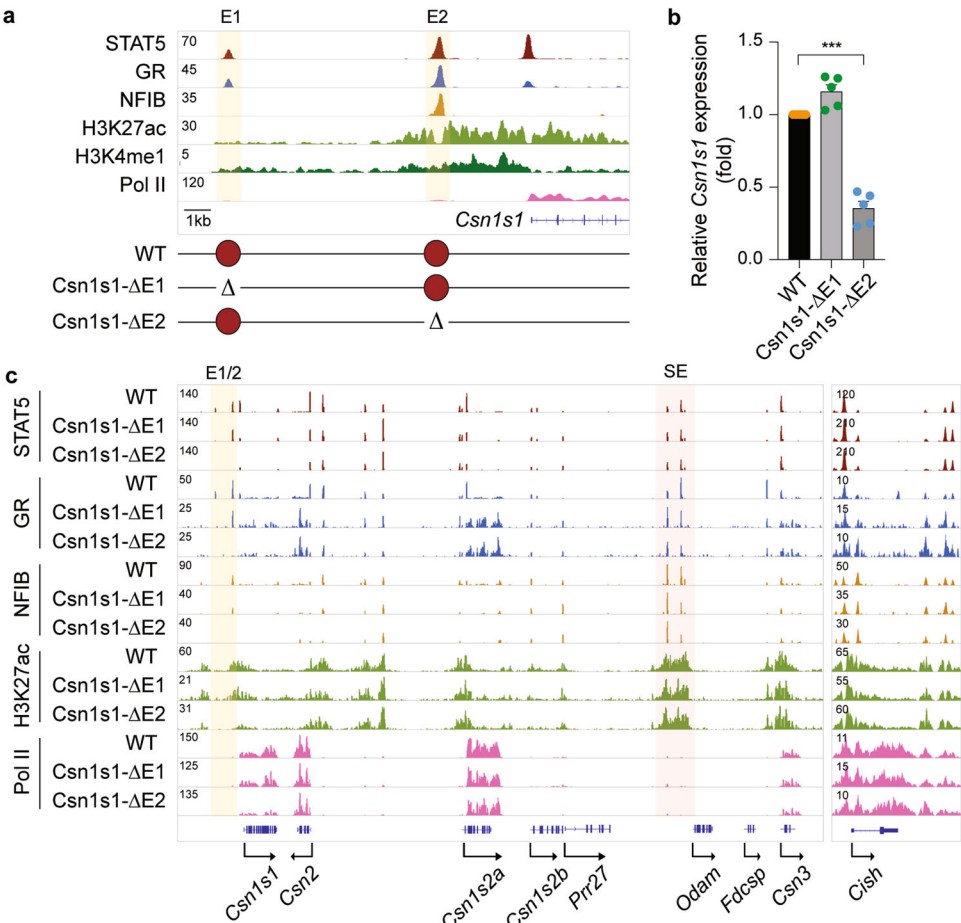

**Fig. 6 | *Csn1s1* gene expression is modulated by a proximal enhancer. a** The candidate *Csn1s1* enhancers were identified by ChIP-seq for TFs and activating histone marks at day one of lactation (L1). The yellow shade indicates the enhancers. The diagram shows the deletions introduced in the mouse genome using CRISPR/Cas9 genome engineering. Red circles indicate *Csn1s1* enhancers. **b** *Csn1s1* mRNA levels in lactating mammary tissues (day one of lactation, L1) from WT and mutant mice were measured by qRT-PCR and normalized to *Gapdh* levels. Results are shown as the means ± SEM of independent biological replicates (*n* = 5). Two-tailed *t*-test with Welch's correction was used to evaluate the statistical significance of differences between WT and each mutant mouse line. ***\*\*\*p* < 0.0001. **c** The *Casein* locus was profiled using ChIP-seq in WT and Csn1s1-E1 and E2 mutant tissues at day one of lactation (L1). The red shade indicates the super-enhancer (SE). The *Cish* locus served as ChIP-seq control. Source data are provided as a Source Data file.

local enhancers essential for their expression in lactating mammary tissue, they still retained their dependence on the SE.

SEs are structurally defined compound regulatory elements which control lineage-specific genes[33]. There is ongoing research investigating if individual enhancer modules have unique properties and their contributions to SE activity. A few studies have tackled the complexity of SEs in mice and gained insight into the biological significance of individual components. The well-studied α-globin SE consists of five individual enhancers that act independently and in an additive fashion[28]. In a recent study, the Higgs group rebuilt the α-globin SE from individual modules and identified elements that were non-functional on their own but facilitated the activities of others[34]. Like the α-globin locus, the casein SE and the Wap SE[16] are composed of modules with distinct transcriptional capacities. Studies from the Higgs group have also demonstrated, unexpectedly, that the α-globin SE activity is orientation-dependent[35] and inversion preferentially activates genes 5′ to the SE. Similarly, the casein SE investigated in our study differentially activates genes located at either end. Based on current knowledge, the regulatory logic of SE is based on mixtures of essential and dispensable modules that can be active in one or more cell types[16,34–41]. Recent studies have demonstrated that enhancers and promoters can engage CTCF-associated sites and induce distant genes[42,43], a concept that might also be applicable in the casein locus,

which harbors four sites occupied by CTCF (Fig. 1a), two bordering the locus and two within the SE[17]. We have deleted these sites from the mouse genome and observed only a moderate effect of the SE-bound sites[17].

There is limited knowledge about regulatory mechanisms operative in salivary tissue. Genome-wide histone modification studies in mouse submandibular glands have pointed to putative regulatory regions[44], but no defined salivary-specific enhancers have been described. ATAC-seq data from salivary tissue have been published[45], but they did not yield additional information. Key TFs controlling mammary function, such as STAT5[8,9] and ELF5[7], are dispensable in the salivary gland[46]. The presence of the progesterone receptor is also required for normal mammary gland development[47], but its role in salivary physiology remains to be determined[48]. The molecular backbone of the casein SE, as defined by H3K27ac marks, extends over 10 kbp, and no salivary-specific TF sites have been identified. While two enhancer modules within the casein SE are key activators in mammary tissue, it is not clear which of the four enhancer elements control salivary gene expression. We also need to consider the possibility that the casein SE activity is facilitated by receptive target gene promoters as indicated by the neighboring *Fdcsp* gene, which is active in salivary but not in mammary tissue. This specificity is not influenced by the distance of the SE or the presence of the intervening *Odam* gene,

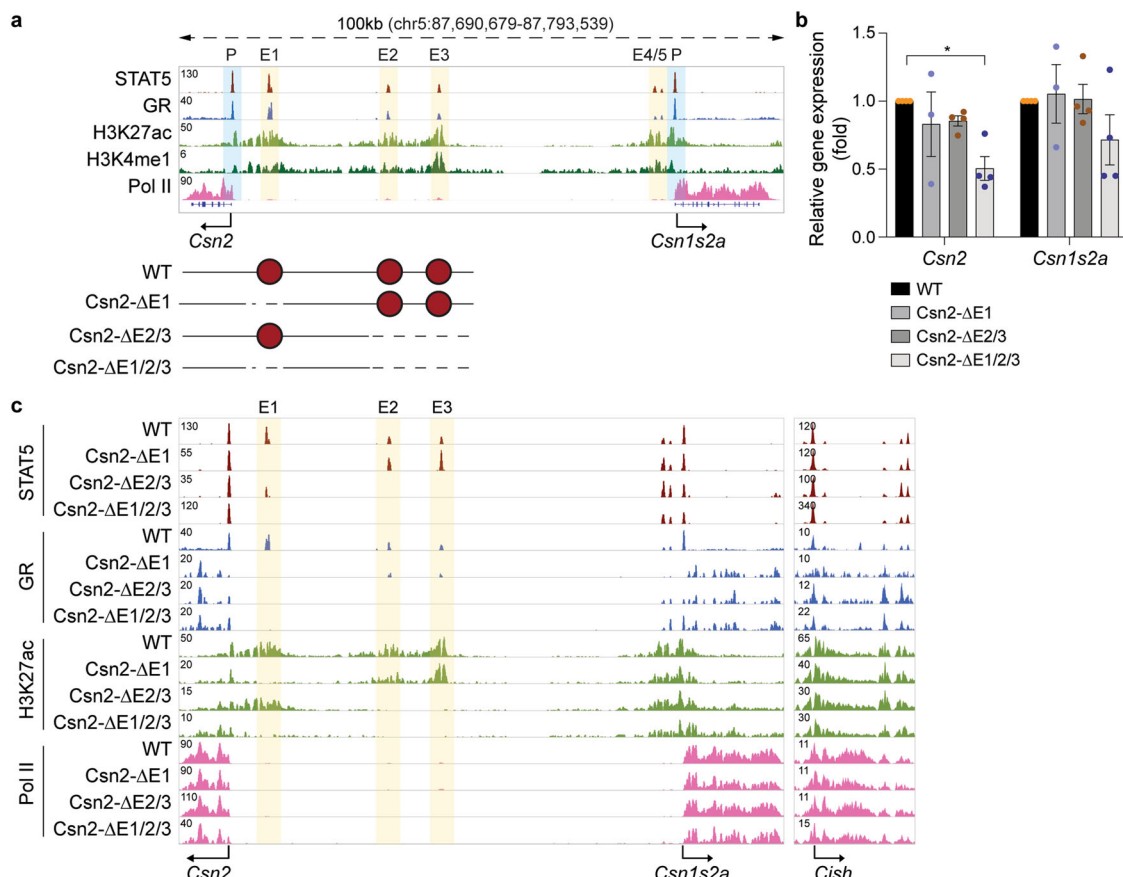

**Fig. 7 | Activity of putative *Csn2* enhancers. a** Chromatin features of the *Csn2-Csn1s2a* locus were mapped by ChIP-seq for mammary transcription factors (TF) and activating histone marks at day one of lactation (L1). The presence of H3K27ac and H3K4me1 marks indicated three candidate enhancers, E1 at −6 kb, E2 at −25 kb and E3 at −35 kb. The yellow and blue shades indicate the enhancers and promoters, respectively. The diagram shows the enhancer deletions introduced in mice using CRISPR/Cas9 genome engineering. Red circles indicate *Csn2* enhancers. **b** Expression of the *Csn2* gene was measured in lactating mammary tissue (day one

of lactation, L1) from WT and mutant mice carrying enhancer deletions by qRT-PCR and normalized to *Gapdh* levels. Results are shown as the means ± SEM of independent biological replicates (WT, ΔE2/3, ΔE1/2/3, *n* = 4; ΔE1, *n* = 3). 2-way ANOVA followed by Dunnett's multiple comparisons test was used to evaluate the statistical significance of differences between WT and each mutant mouse line. *$p < 0.05$. **c** ChIP-seq analysis shows the chromatin structure of the *Csn2* locus in lactating mammary tissue (day one of lactation, L1) of WT and mutant mice. The *Cish* locus served as ChIP-seq control. Source data are provided as a Source Data file.

suggesting the presence of unique promoter elements that permit enhancer sensing. Alternatively, differential promoter accessibility[49] in salivary and mammary tissue could account for cell specificity.

The five *Casein* genes, namely *Csn1s1*, *Csn2*, *Csn1s2a*, *Csn1s2b* and *Csn3*, share common regulatory mechanisms. These include lactogenic hormone response elements, such as GAS motifs bound by the prolactin-activated transcription factor STAT5, glucocorticoid response elements (GREs) and sites recognized by NFIB, another key TF to the 10,000-fold stimulation of casein gene expression during pregnancy and lactation[8,9,16,50]. While there are shared regulatory mechanisms for the five *Casein* genes, the presence of distinct differences provides flexibility and specificity in the regulation of each gene's expression and ultimately contributes to the complexity of milk production. Most notably, the presence of the super-enhancer within the regulatory landscape of *Csn1s2b* and *Csn3* suggests that these genes rely on a highly cooperative and robust network, which includes local enhancers, for their expression. On the other hand, *Csn1s1*, *Csn2* and *Csn1s2a* are largely regulated by a different set of local gene-specific enhancers with minimal dependence on the super-enhancer.

The combination of local gene-specific enhancers and promoters, all bound by STAT5 and other mammary-centric TFs, is likely the regulatory key to casein gene expression. For example, *Csn2* expression is under the combined control of three distal enhancers and a single promoter element, with their individual ablation having a very

limited biological impact. The neighboring *Csn1s1* gene also features two STAT5-bound enhancer elements, and while their individual deletion had a limited biological impact, it can be hypothesized that, like in the *Csn2* gene, the enhancer and promoter elements synergize to ensure maximum transcriptional control.

Although STAT5 is the key switch activating *Casein* genes during pregnancy and lactation, the presence of additional control elements within the promoter and enhancer regions might modulate their activation. Both the *Csn1s2b* and *Csn3* genes display functional and structural features not seen in other *Casein* genes. *Csn1s2b* contains a unique intronic enhancer bound by STAT5 and other mammary TFs that contribute to the priming of the single distal enhancer and the promoter prior to parturition, resulting in a distinctive temporal expression distinct from that of the other *Casein* genes[25]. Lastly, while four *Casein* genes (*Csn1s1*, *Csn2*, *Csn1s2a* and *Csn1s2b*) are characterized by STAT5-bound promoter regions, the *Csn3* promoter does not feature a GAS motif, nor does it bind STAT5 or other mammary-centric TFs. This is probably not surprising since *Csn3* is not a classical calcium-sensitive casein gene and has a distinct evolutionary history.

At least 13 STAT5-based enhancers, including the four component super-enhancer, and five distinct promoters control the five *Casein* genes, a testimony to the regulatory complexity of different building blocks, thus ensuring the maximum production of milk needed for the survival of the young and the species. Although all enhancers are

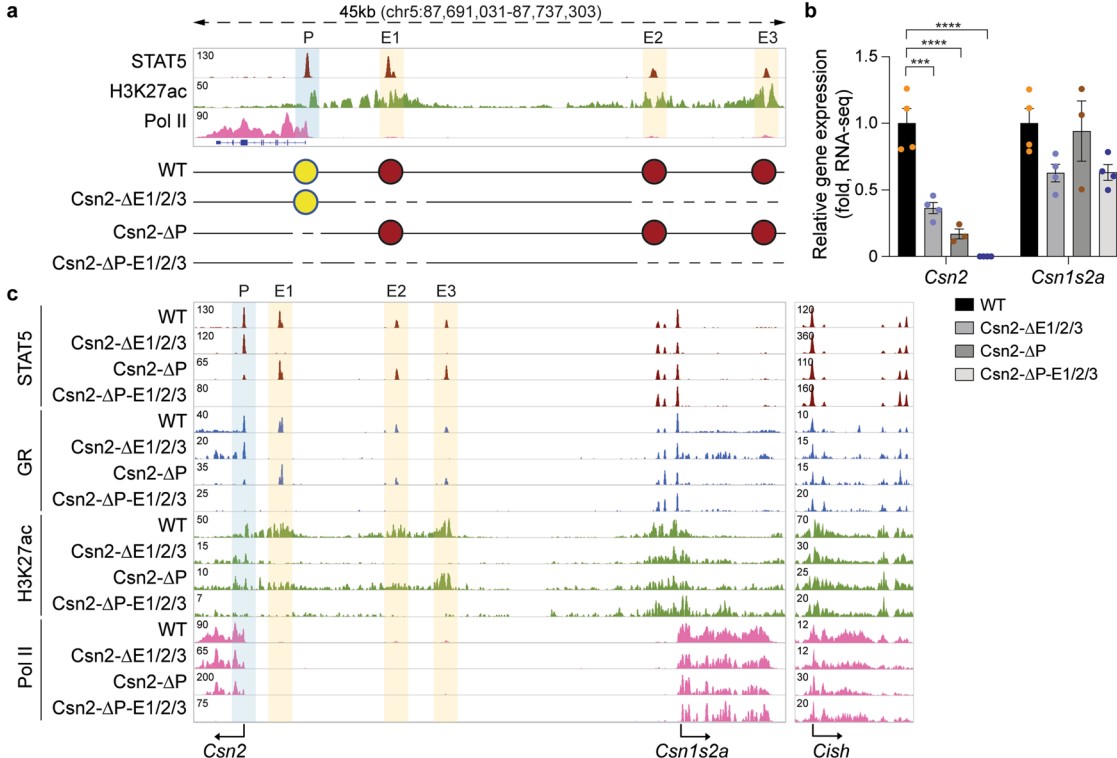

**Fig. 8 | Synergy between promoter-based cytokine-response elements and distal enhancers. a** The *Csn2* promoter region was characterized through ChIP-seq for STAT5, activating histone marks and Pol II loading at day one of lactation. The yellow and blue shades indicate the enhancers and promoters, respectively. The diagram shows the enhancer deletions and promoter mutations introduced in the mouse genome using CRISPR/Cas9 genome engineering and deaminase base editing, respectively. Red and yellow circles indicate *Csn2* enhancers and promoters. **b** *Csn2* mRNA levels were measured by RNA-seq in lactating mammary tissue (day one of lactation, L1) isolated from WT mice and mice carrying disabling

mutations in the two GAS motifs in *Csn2* promoter (ΔP) in the presence and absence of the three distal enhancers (ΔE1/2/3) (WT, ΔE1/2/3, ΔP-E1/2/3, n = 4; ΔP, n = 3). Results are shown as the means ± SEM of independent biological replicates. 2-way ANOVA followed by Dunnett's multiple comparisons test was used to evaluate the statistical significance of differences between WT and each mutant mouse line. ***$p < 0.0001$, ****$p < 0.00001$. **c** Chromatin features of the *Csn2* locus were investigated by ChIP-seq in lactating mammary tissue (day one of lactation, L1) of WT, ΔE1/2/3, ΔP and ΔP-E1/2/3 mice. The *Cish* locus served as ChIP-seq control. Source data are provided as a Source Data file.

bound by STAT5 and an array of mammary TFs, only a subset displays a classical STAT5 binding motif, suggesting that a seemingly equivalent TF occupancy can be achieved through a combination of TFs, including GR and NFIB.

While the *Wap* gene relies on a tripartite SE for its activation, the *Csn2* gene is a unique and distinct example of synergy between distal enhancer elements and promoter elements, both of which respond to cytokines through the mammary TF STAT5. A new class of regulatory elements, named Epromoters[51,52], function as both promoter and enhancer. Epromoters are hubs for TF machinery and are typically associated with stress-response genes, such as those activated by interferons (IFN) through the TFs STAT1/2. A key feature of Epromoters is their capacity to activate neighboring genes independent of the presence of additional enhancers[52]. Epromoters are frequently found in loci harboring co-regulated genes, such as the *Oas* locus, that otherwise do not contain enhancers[52].

Despite similarities between Epromoters and the *Csn2* promoter, there are distinctive differences (Fig. 9c). Although both Epromoters and the five *Casein* gene promoters, including *Csn2* investigated in this study, harbor cytokine-response elements bound by TF STATs, their positioning appears to be different, with those in the *Casein* genes being within 100 bp of the TSS and therefore likely integral part of the promoter. A distinct difference between the two regulatory elements is the capacity of Epromoters to activate neighboring genes in the absence of additional enhancers, while the *Csn2* promoter element is fully dependent on the presence of distal enhancers. Moreover, unlike Epromoters, which activate neighboring genes at great distance, the

*Csn2* promoter activity is confined to its own gene, and neighboring genes are regulated independently. The activity of Epromoters has been validated in cell lines using CRISPR-Cas9-induced deletions spanning several hundred base pairs[52], which leaves the possibility that additional elements contributed to their activity. In contrast, our study introduced single base pair mutations into GAS motifs, demonstrating the specific and defining requirement of STAT5 in the activation of *Csn2*. Validating the specificity of TF binding through the introduction of point mutations is desirable since it has been shown that phantom TF binding occurs at active promoters[53].

Regulation of the *Csn2* gene has been investigated in HC11 tissue culture cells by Kabotyanski and Rosen[54,55]. Prolactin treatment resulted in the recruitment of STAT5 to the promoter region and a candidate enhancer at −6 kb[54], and looping between these elements was observed[55]. Efficient RNA polymerase II recruitment required the simultaneous administration of prolactin and glucocorticoids[54], which is supported by our study demonstrating STAT5 and GR binding to these elements. While the −6.4 kb enhancer identified by the Rosen group coincides with the Csn2-E1 enhancer in our study, Csn2-E2 and Csn2-E3 have not been reported in HC11 cells, pointing to clear differences between in vivo mouse studies and tissue culture cells. A study using cell culture and gene transfection chloramphenicol acetyl transferase (CAT) assays as readout described an element in the upstream region of the bovine beta *Casein* gene (now *Csn2*) that responds to the extracellular matrix and prolactin[56,57]. Another investigation also using gene reporter assay described a similar element in the human beta casein gene[58]. While these elements activate reporter

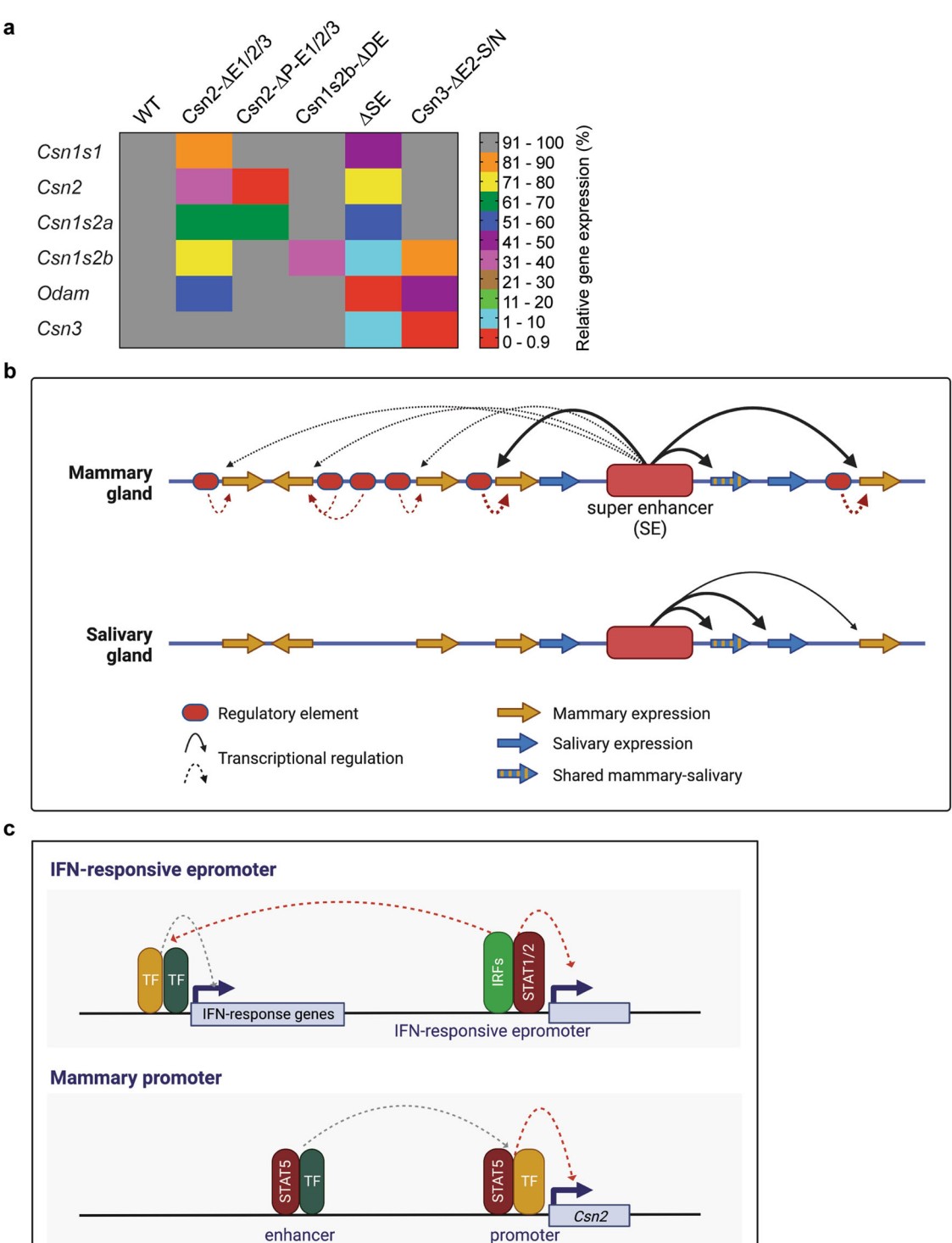

**Fig. 9 | Model for the regulation of the shared casein mammary-salivary locus by a dual specific super-enhancer, gene-specific local enhancers and promoters. a** Heatmap shows relative gene expression levels of genes in the *Casein* locus in the various mutant mice compared to WT mice at day 18 of pregnancy (p18) or day one of lactation (L1). DE distal enhancer. The data from the *Csn2, Csn3* and ΔSE mice are from this study, and the Csn1s2b mutant has been reported earlier[25]. **b** The super-enhancer preferentially activates the *Csn1s2b, Csn3* and *Odam* genes and marginally the *Csn1s1, Csn2* and *Csn1s2a* genes in mammary gland during pregnancy and regulates the promoters of *Odam, Fdcsp* and *Csn3* genes in salivary gland tissue. **c** The *Csn2* gene is characterized by distinct enhancer and promoter elements, both harboring STAT5 binding sites (GAS motifs) that are bound by STAT5 during lactation. Individually, neither the enhancers nor the promoter

STAT5 sites are absolutely required for efficient gene activation, suggesting that they can partially compensate for each other's activity. Combined inactivation of the distal enhancers and the promoter STAT5 site completely abrogates *Csn2* expression suggesting synergism during pregnancy and lactation. Also, neighboring mammary genes are not impacted by the *Csn2* enhancer and promoter elements. This distinguishes them from Epromoters[51,52] that bind STAT TFs and have been identified in interferon-response genes. In addition to their own native gene, Epromoters also control the expression of neighboring genes. Neither Epromoter-associated genes nor neighboring genes harbor classical distal enhancers. The graphs were created with BioRender.com. Source data are provided as a Source Data file.

genes in cell lines, deletion of the corresponding elements in the mouse casein locus has limited significance, as shown in our in vivo study, emphasizing clear differences between cell culture studies and in vivo mouse genetic studies.

The *Csn2* promoter appears to be a hub for transcription factors, such as the progesterone receptor[59] and C/EBPs[60]. Progesterone suppresses *Csn2* expression[61], and the receptor is essential for mammary gland development[48] suggesting a wider role in the mammary genome. As the complexity of pregnancy-induced genetic programs and the magnitude of gene activation cannot be replicated in tissue culture cells, experimental mouse genetics provides an opportunity to gain insight into physiological processes. Having said this, global or cell-specific deletions of regulatory proteins, such as transcription factors, frequently elicit global effects, which prevents a clear understanding of their roles in individual genes. The inactivation of specific regulatory elements through genome engineering provides more definitive answers.

Gene families regulated by IFNs, and milk protein genes share features that likely co-evolved during evolution. Repurposing structural genes, promoters and enhancers is a driving force in evolution, fostering innovation and the establishment of new genes and regulatory concepts[62,63]. Both, IFN-regulated genes and milk protein genes, are rapidly induced by cytokines that specifically utilize the JAK/STAT regulatory machinery permitting a rapid induction of genes. However, while transcriptional activation elicited by IFNs largely relies on JAK1 and STAT1/2[64], mammary genes are predominantly activated by the pregnancy hormone prolactin, signaling through JAK2 and STAT5[8,9,11]. The elaborate enhancer structure in the casein locus permits an exceptional expression of five genes accounting for 80% of milk proteins, an essential requirement for the sustained success of mammals. We propose that the compactness of enhancers and their high-density occupation with TFs and co-activators provides an optimal regulatory environment. In summary, the comprehensive dissection of promoter and enhancer elements within their native loci provided physiological insight into a complex hormone-controlled regulatory network operative in mammary and salivary tissues.

# Methods

## Mice

All animals were housed in an environmentally controlled room (22–24 °C, with 50 ± 5% humidity and 12 h/12 h light–dark cycle) and handled according to the Guide for the Care and Use of Laboratory Animals (8th edition) and all animal experiments were approved by the Animal Care and Use Committee (ACUC) of National *Institute* of Diabetes and Digestive and Kidney Diseases (NIDDK, MD) and performed under the NIDDK animal protocol K089-LGP-17. CRISPR/Cas9 targeted mice were generated using C57BL/6N mice (Charles River) by the transgenic core of the National Heart, Lung, and Blood Institute (NHLBI). Single-guide RNAs (sgRNA) were obtained from either OriGene (Rockville, MD) or Thermo Fisher Scientific (Supplementary Table 3). Target-specific sgRNAs and in vitro transcribed *Cas9* and *Base editor* mRNA were co-microinjected into the cytoplasm of fertilized eggs for founder mouse production[65–68]. The ΔE1/2 mutant mouse was generated by injecting a sgRNA for E2 into zygotes collected from ΔE1 mutant mice. The Csn2-ΔE1/2/3 and Csn2-ΔP-E1/2/3 mutant mice were generated by injecting a sgRNA for E1 into zygotes collected from Csn2-ΔE2/3 mutant mice and for P into zygotes collected from Csn2-ΔE1/2/3 mutant mice, respectively. All mice were genotyped by PCR amplification and Sanger sequencing (Macrogen and Quintara Biosciences) with genomic DNA from mouse tails (Supplementary Table 4).

Two-months old mice were used in the experiments. Mammary tissue was collected from non-parous (virgin) mice, from mice at day 6 of pregnancy and at days 1 (L1) and 10 (L10) of lactation and stored at −80 °C. While most mutations did not affect the ability of dams to lactate and nurse their pups, deletion of the super-enhancer (ΔSE and ΔE1/2) and deletion of the *Csn3* E2 enhancer (Csn3-ΔE2) resulted in lactation failure, which is most likely due to the absence of CSN3. The C57BL/6 mouse strain used in this study has a gestation period of 18.5 days[21], and mammary tissues from mutant mice unable to lactate (ΔSE, SE-ΔE1/2 *and* Csn3-ΔE2) were collected at day 18 of pregnancy (p18) or within 12 h after delivery (post-partum <12 h, pp <12 h) (Supplementary Data 10). Salivary tissues (submandibular glands) were collected from 2-month-old male mice and stored at −80 °C.

## Chromatin immunoprecipitation sequencing (ChIP-seq) and data analysis

The frozen-stored tissues were ground into powder in liquid nitrogen. Chromatin was fixed with formaldehyde (1% final concentration) for 15 min at room temperature and then quenched with glycine (0.125 M final concentration). Nuclei were isolated with Farnham Lysis Buffer (5 mM PIPES pH 8.0, 85 mM KCl, 0.5% NP-40, PMSF and proteinase inhibitor cocktails). The chromatin was fragmented to 200–500 bp using sonicator 3000 (30 cycles; 20 s pulse/20 s rest, Misonix Sonicators) and further lysed in RIPA buffer. One milligram of chromatin was immunoprecipitated with Dynabeads Protein A (Novex) coated with antibodies. The following antibodies were used for ChIP-seq: STAT5A (Santa Cruz Biotechnology, sc-271542), GR (Thermo Fisher Scientific, PA1-511A), NFIB (Sigma-Aldrich, HPA003956), MED1 (Bethyl Laboratory, A300−793A), H3K27ac (Abcam, ab4729), RNA polymerase II (Abcam, ab5408), H3K4me1 (Active Motif, 39297) and H3K4me3 (Millipore, 07-473). Then, 5–10 ug of antibodies were added to 1 mg of total proteins (1 ml solution). After serial bead washes, ChIP DNA was reverse crosslinked at 65 °C overnight in the presence of 1% SDS and 1 mg/ml of Proteinase K (Invitrogen), and DNA was purified with QIAquick PCR Purification Kit (Qiagen). The DNA fragments were blunt-ended and ligated to the Illumina index using the NEBNext Ultra II DNA Library Prep kit for Illumina (New England BioLabs). Libraries for next-generation sequencing were prepared and sequenced with a HiSeq 3000 instrument (Illumina).

Quality filtering and alignment of the raw reads were done using FastQC (version 0.11.0), Trimmomatic[69] (version 0.36) and Bowtie[70] (version 1.2.2), with the parameter '-m 1' to keep only uniquely mapped reads, using the reference genome mm10. Samtools[71] was to convert BAM files to SAM format. Picard tools (Broad Institute. Picard, http://broadinstitute.github.io/picard/. 2016) was used to remove duplicates, and subsequently, Homer[72] (version 4.9.1) and deepTools[73] (version 3.1.3) software were applied to generate bedGraph files and normalize coverage, separately. Integrative Genomics Viewer[74] (version 2.3.98) was used for visualization. Coverage plots were generated using Homer[72] software with the bedGraph from deepTools as output. R and the packages dplyr (https://CRAN.R-project.org/package=dplyr) and ggplot2[75] were used for visualization. Each ChIP-seq experiment was conducted for two replicates, and the correlation between the replicates was computed by Spearman correlation using deepTools (Supplementary Data 10).

Although we would ideally want to study the absolute levels of binding, comparing the levels of ChIP-seq enrichments across different conditions is more difficult than one would have hoped for because particularly ChIP-seq is awfully noisy and dependent on too many experimental parameters[76–78]. The quantification of ChIP-seq varies due to experimental variation between samples introduced by different efficiencies in nuclear extraction, DNA shearing and immunoprecipitation. However, the methods for normalization to minimize sample-to-sample variability between conditions are based on the assumption that experimental variables remain constant between datasets and assume comparable genomic binding of the protein between conditions. To overcome this issue, we used deepTools and its bamCoverage function to calculate the coverage as the number of reads per bin and normalize data by Reads Per Kilobase per Million

mapped reads (RPKM) and 1x depth (reads per genome coverage, RPGC). Additionally, we normalized each peak against sequencing depth again. However, these analytical normalization methods didn't normalize experimental variation between conditions[79], even though we generated the replicates. To overcome this issue, we presented the entire casein locus and the *Cish* locus as a ChIP-seq control to show the ChIP-seq pattern.

### Total RNA sequencing (Total RNA-seq) and data analysis
Total RNA was extracted from frozen mammary and salivary tissue from wild-type and mutant mice and purified with RNeasy Plus Mini Kit (Qiagen, 74134). Ribosomal RNA was removed from 1 µg of total RNAs, and cDNA was synthesized using SuperScript III (Invitrogen). Libraries for sequencing were prepared according to the manufacturer's instructions with TruSeq Stranded Total RNA Library Prep Kit with Ribo-Zero Gold (Illumina, RS-122-2301), and paired-end sequencing was done with a HiSeq 3000 instrument (Illumina).

Total RNA-seq read quality control was done using FastQC (version 0.11.0), Trimmomatic[69] (version 0.36) and STAR RNA-seq[80] (version STAR 2.5.4a) using paired-end mode was used to align the reads (mm10). Samtools[71] was to convert BAM files to SAM format and HTSeq[81] was to retrieve the raw counts, and subsequently, R (https://www.R-project.org/), Bioconductor[82] and DESeq2[75] were used. Additionally, the RUVSeq[83] package was applied to remove confounding factors. The data were pre-filtered keeping only those genes, which have at least ten reads in total. Genes were categorized as significantly differentially expressed with an adjusted p-value below 0.05 and a fold change > 2 for upregulated genes and a fold change of <−2 for downregulated ones. The visualization was done using dplyr (https://CRAN.R-project.org/package=dplyr) and ggplot2[84].

### Bisulfite-seq and data analysis
In this, 200 ng of purified genomic DNA was used to create bisulfite sequencing libraries via the large DNA insert approach using the NEBNext® Enzymatic Methyl-seq Kit (New England Biolabs). In short, purified genomic DNA was fragmented, adenylated, and ligated to EM-seq adapters. Bisulfite conversion was performed according to the manufacturer's instructions. Internal unmethylated lambda DNA and CpG methylated pUC19 DNA controls were included with each library. Libraries were sequenced with a NovaSeq 6000 instrument (Illumina) to generate a minimum of 300 million paired-end 151 base reads per library.

Raw fastq files were quality trimmed using TrimGalore (version 0.6.7). Reads were aligned to the mouse genome (mm10) using Bismark[85] (version 0.23.0). Mapped BAM files were sorted by query name, deduplicated using Picard (version 2.27.3) and finally sorted coordinates using Samtools[86] (version 1.17). Integrative Genomics Viewer[74] (version 2.5.3) was used for visualization.

### RNA isolation and quantitative real-time PCR (qRT-PCR)
Total RNA was extracted from frozen mammary tissue of wild-type and mutant mice using a homogenizer and the PureLink RNA Mini kit according to the manufacturer's instructions (Thermo Fisher Scientific). Total RNA (1 µg) was reverse transcribed for 50 min at 50 °C using 50 µM oligo dT and 2 µl of SuperScript III (Thermo Fisher Scientific) in a 20 µl reaction. Quantitative real-time PCR (qRT-PCR) was performed using TaqMan probes (*Csn1s1*, Mm01160593_m1; *Csn2*, Mm04207885_m1; *Csn1s2a*, Mm00839343_m1; *Csn1s2b*, Mm00839674_m1; *Odam*, Mm02581573_m1; *Csn3*, Mm02581554_m1; mouse *Gapdh*, Mm99999915_g1, Thermo Fisher Scientific) on the CFX384 Real-Time PCR Detection System (Bio-Rad) according to the manufacturer's instructions. PCR conditions were 95 °C for 30 s, 95 °C for 15 s, and 60 °C for 30 s for 40 cycles. All reactions were done in triplicate and normalized to the housekeeping gene *Gapdh*. Relative differences in PCR results were calculated using the comparative cycle threshold ($C_T$) method and normalized to *Gapdh* levels.

### Identification of regulatory elements bound by STAT5 in lactation
MACS2[87] peak finding algorithm was used to identify regions of ChIP-seq enrichment over the background to get regulatory elements at L1 and L10. Peak calling was done on both STAT5A replicates and broad peak calling on H3K27ac. Only those peaks were used, which were identified in both replicates and with H3K27ac coverage underneath.

### Identification of complex mammary loci
Mammary preferential genes were identified using RNA-seq data from pregnancy day six (p6), lactation day one (L1) and ten (L10). Those genes were considered, which were induced more than two-fold with an adjusted p-value below 0.05 between p6 and L1 or p6 and L10. The next step comprised the stitching of neighboring genes, by only considering protein-coding genes. Those stitched loci were subsequently compared to the contact domains (Hi-C data), and only those loci passed the validation that were not overlapping with any border of the contact domains. If a locus overlapped, they were treated the following way: (1) if the locus contains only two genes, it was discarded, as it will not pass the prerequisite that a complex locus comprises at least two genes; (2) all other loci were split up at the border, and only those were kept that comprised more than two genes; (3) the loci were shrunk to the size of the remaining genes. As possible regulatory elements (in our analysis STAT5A) are also part of complex loci, we expanded the borders of each locus to comprise STAT5A binding sites, if they were located within the adjacent intergenic region. Those new loci were finally checked again to not overlap with contact domain boundaries, otherwise, they were shrunk down to the last element not overlapping with the contact domain; (4) the final list of complex loci comprises loci with at least three genes.

The analysis was done using bedtools[88], bedops, R (https://www.R-project.org/) and Bioconductor[82], as well as the R packages dyplr (https://CRAN.R-project.org/package=dplyr) and ggplot2[84].

### Statistics and reproducibility
All samples that were used for qRT-PCR and RNA-seq were randomly selected, and blinding was not applied. No data were excluded from the analyses. Differential expression gene (DEG) identification of RNA-seq data used Bioconductor package DESeq2 in R. *P*-values were calculated using a paired, two-side Wilcoxon test and adjusted *p*-value (pAdj) corrected using the Benjamini–Hochberg method. The cut-off value for the false discovery rate was pAdj > 0.05. For comparison of samples, data were presented as the standard deviation in each group and were evaluated with a *t*-test and 2-way ANOVA multiple comparisons using PRISM GraphPad. Statistical significance was obtained by comparing the measures from the wild-type or control group, and each mutant group. A value of \*$P < 0.05$, \*\*$P < 0.001$, \*\*\*$P < 0.0001$, \*\*\*\*$P < 0.00001$ was considered statistically significant; ns, not significant.

### Reporting summary
Further information on research design is available in the Nature Portfolio Reporting Summary linked to this article.

## Data availability
The ChIP-seq, RNA-seq, and Bisulfite-seq data generated in this study have been deposited in the Gene Expression Omnibus (GEO) database under accession code GSE231441. Other data were obtained from the GEO database under accession code GSE25105[22,89], GSE67386[23], GSE74826[16], GSE115370[29], GSE127144[25], and GSE161620[25]. The datasets used in the study are listed in Supplementary Data 10. The fastq files and the processed bedGraph files can be downloaded from GEO

(https://www.ncbi.nlm.nih.gov/gds/?term=) and imported into the IGV browser (https://software.broadinstitute.org/software/igv/download) with a reference genome (mm10). The RNA-seq data generated in this study are provided in the Supplementary Data files. The remaining data generated in this study are provided in the Source Data file. Source data are provided with this paper.

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

## Acknowledgements

We thank Ilhan Akan, Sijung Yun and Harold Smith from the NIDDK genomics core for NGS and Hernan Lorenzi from the NIDDK TriLab core for advice in bioinformatic analysis. This work utilized the computational resources of the NIH HPC Biowulf cluster (http://hpc.nih.gov). H.K.L., M.W. and L.H. were supported by the Intramural Research Programs (IRPs) of the National Institute of Diabetes and Digestive and Kidney Diseases (NIDDK), and C.L. was supported by Intramural Research Programs (IRPs) of National Heart, Lung, and Blood Institute (NHLBI).

## Author contributions

H.K.L. and L.H. designed the study and analyzed data. C.L. generated mutant mice. H.K.L. established mutant mouse line. H.K.L. performed experiments and data analysis. M.W. identified multigene loci. H.K.L. and L.H. supervised the study. H.K.L. and L.H. wrote the manuscript, and all authors approved the final version.

## Funding

## Competing interests

The authors declare no competing interests.
