## [Peer Review File · Nature Communications]

Cell-specific and shared regulatory elements control a multi-gene locus active in mammary and salivary glandsREVIEWER COMMENTS

Reviewer #1 (Remarks to the Author):

Summary:

In the manuscript titled “Cell-specific and shared enhancers control a high-density multi-gene locus active in mammary and salivary glands”, Lee et al. have investigated the role of enhancers and super-enhancers (SE) in the regulation of genes present in a specific locus that has relevant roles in both mammary- and salivary-gland tissues. The most interesting aspect of this study is the choice of locus they have chosen to study given both its complexity – it contains 8 genes and 20 candidate regulatory elements – and its mutually shared as well as exclusive activity in the 2 distinct tissues. The overall approach of employing experimental mouse genetics for this study is also sound and frequently yields intriguing results. However, significant concerns regarding (a) the rationale and (b) interpretation of those results temper the conclusions claimed by the authors in this manuscript. Additionally, several minor transgressions in the writing also accumulate to form a major concern regarding the inferences supporting the conclusions of the study.

Major concerns:

1. The authors need to be consistent in the organization and presentation of related data of their figures. For example, in different instances, the authors display comparative RNA-Seq results as “relative fold of normalized read counts” and “relative gene expression (fold)”, “normalized read counts (log10)” instead of using one standard metric for such comparisons. Such choices raise additional questions regarding the methods used to make these comparisons, that are neither addressed in the figure legends nor in the methods section.
2. Genome tracks do not have comparative scales. Since the height of the peaks depend on sequencing depth and transcript abundance, we understand (with some reservations) the choice of using different scales to simply compare the presence and absence of peaks in the WT and the experimental (deletions) track. However, this approach has a major pitfall when comparing the different deletions. For example (but applies to almost every figure), in supplementary figure 4, STAT5 binding in $\Delta E2/E4$ is significantly increased at E1 (160 vs 75 on y-axis) – does it indicate compensatory binding at E1 when E2 and E4 are not available? What does it mean for the expression and regulation of the genes under the control of the enhancer? The author must consider these questions, as they’re fundamental to the understanding of gene regulation at this locus. Additionally, the H3K27ac peaks at E1 for $\Delta E1$ (absent) and $\Delta E2$ (present) could be quite comparable since the low peaks in $\Delta E1$ are scaled to 60 and in $\Delta E2$ are scaled to 10. The authors might be aiming to compare the relative enrichment of peaks in E1 and E4, but this approach is ultimately biased and we must recommend extreme caution when making inferences from such results.
3. In several figures involving genome tracks, why do we observe any peaks in the locus that are specifically deleted in the ChIP-Seq samples? This is likely background, but raises the question of why the authors haven’t normalized the samples with, for example, an IgG control?
4. The authors provide a very skewed perspective on the alleged evolutionary changes observed in the locus – from their writing, they seem to indicate that the high-density multi-gene locus evolved with the goal of somehow enhancing the regulatory capabilities of the various elements that resulted from events such as gene duplication etc. and is not merely a characteristic that is more adaptive in nature. To adequately support their claims, the authors must also employ other species in their experimental design, even if to merely address it from a genomic perspective.

Minor concerns:

1. The authors have used fold-change (log-normalized or otherwise) to show relative mRNA enrichment in comparisons of different samples, then they should be consistent in their text by reporting relative expression changes in fold-changes and not %-ages.
2. Through the writing, the authors must attempt to make the paper more accessible to a broader scientific audience. The writing is rife with jargons that are specific to the niche community (for example, p6 can mean “pregnancy day 6” as well as “post-natal day 6” to different group of readers) and the authors must make the effort to define such specific terms in the main text or the methods.
3. The authors must improve the language of the manuscript more generally. Each paragraph must have a clear rationale and proper conclusion to ensure flow of logic, which is largely missing from the manuscript making it a generally difficult read. Specific examples of concerning language include (but are not limited to):
 - a. Page 1, line 30: “...success of milk and saliva” – given that success refers to the outcome of a process, which process involving milk and saliva are the authors referring to, should be made clearer.

- b. Page 1, line 30-31: "...mineralized tissues, like mammary and salivary glands" – examples of mineralized tissues include nails and teeth. Mammary glands and salivary glands are not mineralized tissues.
- c. Page 2, line 61: "...has all the hallmarks of a super-enhancer" – which specific hallmarks are the authors referring to? The increased H3K27ac peaks?
- d. Page 3, line 63: "chromatin structures" – the authors have not assessed structures, but protein occupancy.
- e. Page 4, line 94: "to understand... we conducted additional ChIP-seq experiments" – but with what changes in conditions?

4. Most of the genome tracks do not have consistent labeling, which results in significant confusion on part of the reader.

Reviewer #2 (Remarks to the Author):

In this manuscript the authors describe an extensive study of the regulatory landscape of the casein gene locus which contains genes expressed in epithelial cells, 5 casein genes expressed predominantly and at high levels in the lactating mammary gland epithelium and 3 genes expressed predominantly in the salivary gland epithelium among other tissues, one of which is also detected in the lactating mammary gland. This is a real tour the force with the sheer number of deletion-mutant mice analyzed for different aspects of transcription factor binding, chromatin marks and expression. It gives fascinating insights in the complexity of the regulation of a large multi gene locus predominantly expressed in two different epithelial tissues. It makes clear that both larger Super-enhancers and smaller locally acting enhancers and promoters are important to achieve full functionality, and that some level of redundancy might exist to safeguard a function as important for species survival as lactation. The aspect of temporal regulation in the mammary gland is however not well addressed and could be worked out better especially within the context of previous work by these authors and others and the reported findings in this manuscript related to effects in early pregnancy.

Conceptually, I Strongly suggest including the Csn2-enhancer data reported in the accompanying paper to provide a complete picture especially as this is part of the locus and just as all other casein genes Prl-Jak/stat regulated. I do not think much is gained by reporting this separately and not in the current form. My main problem is that in general, the manuscript fails to contextualize the reported findings with previous work reported by others on regulatory elements, chromatin organization, epigenetic modifications and expression at different physiological time points in the mammary gland. Most of the regulatory regions identified here, are conserved and have been identified before by others in mouse and other mammals and have been studied in the context of chromatin organization and chromatin interactions, gene regulation, and transcription factor binding in the mammary gland and MEC at different physiological states. E.g. CSn1s1 E2 (ECR3 mouse/cow-development/differentiation, Cow -mastitis and involution) CSn2-E1 = BCE (conserved) identified in human, cow and mouse, studied in vitro and in vivo (although this region is not covered in this manuscript but the accompanying manuscript), Mouse ECR19 = SE-E1). The authors fail to mention that the SE mutant mice show lactation failure (mentioned in a previous paper by the authors) even though this can be expected when Csn3 expression is virtually wiped out (DOI: 10.1073/pnas.0601611103), this is not addressed in the manuscript, nor is the expected consequence of the different enhancer deletions and mutations based on results of Csn1s1 and csn2 KO mice discussed. Csn1s1, Csn2 and Csn3 knocked-out mice have been reported with different effects on milk production. This probably explains why expression is analyzed in P18 tissues, this needs to be made clear and discussed. It also would be helpful to discuss the finding in the context of previously reported CTCF work by the authors. What is the role of CTCF in the mammary gland vs SG? And how do the deletions described here correlate to CTCF site/binding and how could this interact/affect CTCF binding/function. At least discuss this. As this work spans assessment of several regulatory elements in the casein locus it would be very helpful to put this in perspective to previous work by others and enables readers to understand the state of the field, and the confirmation of previous work and the new contributions as reported in this study. To aid in reader's understanding add details on which time points (p6, p18, L1, L10) are being analyzed and reported on (in text, figures and figure legends) and why certain time points are selected. Temporal aspects of the regulation should be addressed more thoroughly, as it has been reported that regions encompassed in the SE (SE-E1) as well as Csn1s1-E2 show open chromatin (based on histone marks and DNA hypomethylation) in virgin MEC and virgin MEC subpopulations (Rijkels, 2013 and DosSantos, 2015)

- What is the effect of deleting the SE, CSn1s1-E1 or E2, or Csn3 E1, E2 or P on virgin casein and other locus gene expression as well as expression at the different developmental stages including L1 and L10. The locus undergoes gradual changes in chromatin organization with functional differentiation and milk protein production how is that reflected in the effect of the different enhancer/ promoter mutants?

- IT would be helpful to show Chromatin enhancer markers/ TF binding of the different regulatory elements in the whole locus for the different deletions, as well as expression of all genes in the locus, preferably at the different time points. Currently only subsets of the locus are shown in several cases.
- Showing the coverage trace-difference changes in binding/enrichment using coverage profiles for the different regulatory regions for P6 P18, L1, L10 and WT vs Mut, similar to the coverage plots in Lee et al 2018, will help to clearly visualize changes. The current aggregate coverage plots do not visualize this and the differences in coverage are hard to see in some cases in genome browser view images.
- Based on the binding of the TF analyzed and timing and the effect of deleting certain enhancers. Can you hypothesize which factor(s) aid in opening chromatin in the different enhancers and the timing of it?
- other factors besides STAT5, NF1, GR have been shown to play a role in regulation of genes in this locus (e.g., ELF5, CEBP/B, even PR and YY1) I know not all of these can be studied but some mention is warranted.
- In figure 1 and sup figure 1 instead of showing p6 vs L1 and virgin vs L10 for WT and then showing several p18 WT vs MUT RNA-seq data sets, can you show expression changes in the casein locus genes from Virgin through L10 in one figure?

Other comments

Unconventional manuscript structure: Introductory paragraph, main text, materials and methods. Instead of Abstract, introduction, results, discussion materials and methods.

Use of mammary specific vs preferential expression, be consistent. I suggest preferential expression.

Same nomenclature should be adopted throughout the manuscript (including Materials and Methods) for the enhancers/ promoters that includes the gene name for which the promoter or different enhancers are mutated/ investigated, there are too many E1's and E2's.

In the "introductory paragraph" (abstract?)

It might be useful to mention all genes present in the locus by name, especially because two are mentioned subsequently.

Line 21/22: "the other three caseins" only one of the five has been mentioned >> revise as appropriate by mentioning *csn1s2b* (I assume) or rephrasing. Also make clear when in the mammary gland the 98% reduction takes place.

Line 30/31: salivary gland and mammary gland are not mineralized tissues; they are secretory epithelial tissues and incorporate calcium in their secretion. And the genes in this region encode secretory calcium binding phospho-proteins, contributing in the case of the *Odam* gene to dental mineralization. ref 2 and 3 are appropriate, other references by the same authors specifically addressing the duplication giving rise to the casein locus are more appropriate.

Line 190/191 please provide appropriate references.

Prr27 is not reported on, why?

For the stich analysis, please share the final list of identified complex loci and the genes they contain in a supplementary table.

It seems that the identification of regulatory elements was focused only on those that contain STAT5 binding. Please explain and discuss limitations.

If a piece of DNA is deleted, as is the case in most of the enhancer mutants reported on here, it is expected that you do not detect binding of transcription factors or presence of histone marks, as the DNA is not there to bind or even to be sequenced. The lack of detection of such binding confirms the deletion but nothing else, so pointing out the lack of binding is a bit fraught, especially for TFs binding. It might be worth pointing out the loss of histone marks in the surrounding DNA as this implies that the deleted DNA might play a role in setting up the chromatin mark.

Individual SE enhancer mutations

What happens to the Binding of STAT5, GR, NF1 and chromatin marks at *Csn1s1*, *Csn2*, *Csn1s2a* upon deletion of the individual SE enhancers dE1-4?

DO you see similar changes in expression as for the whole SE deletion at earlier time points?

What happens in the salivary gland with the individual dE1-4 deletions? Can you predict other TF in these regions that could be active in SG?

Does the SE deletion (or Odam deletion) affect expression of salivary and dental epithelial genes located more distal to the casein locus (such as *smr*, *prol* genes or *Ambn*, *Enam*, *Amel*)?

Csn1s1-E1 and -E2 mutations:

The deletion of the GR motif in E1 also leads to loss of GR at the promoter (this is not mentioned), but this does not appear to change expression levels. With respect to the E2 mutation, although it is not clear at what stage of lactation the expression of *Csn1s1* was analyzed (L1, L10?) a 65% reduction in expression is significant in the context of milk production/ nutritional value, if that translates to a reduction in protein content of the milk. If this is at L1 is this more profound at L10 or less so? What happens to expression of the other *Csn* genes and non-*Csn* genes in the locus? It is worth noting that *Csn1s1-E2* is the same as *ECR3* reported by Rijnkels et al (2013), which has an open chromatin conformation in virgin MEC just like *SE-E1* (Rijnkels 2013, DosSantos 2015). As the *SE* deletion shows more profound effects on gene expression at P6 or earlier what happens to the expression of genes in the locus upon mutation of *Csn1s1-E2*?

In the *Csn1s1-E2* mutant it appears that *STAT5* binding is only lost at E1 (and reduced at the promoter), while *GR* binding is lost at E1 and the promoter, which could lead to impaired regulation of *Csn1s1* expression, it should be pointed out in the text. As it is unclear at what stage the ChIP was done, qPCR is lactation (L1 or L10?), does this change at L10? Or at earlier time points? Does the deletion affect *ELF5* binding site too?

General questions related to the generation of deletion mutants via CRISPR/Cas9.

- Can you rule out off target effects? As several of the gRNA target regulatory elements with e.g. *STAT5* sites and many milk protein genes are regulated by the same set of transcription factors.
 - o What happens to chromatin marks and TF binding at other milk protein genes and milk fat synthesis related genes, and what happens to their expression?

- The *SE* deletions and *Odam* deletions are large deletions, 8-10KB, how do they affect 3D organization of the locus? and what is the effect of moving certain enhancers and promoters closer together?

Materials and Methods:

Provide details for salivary gland analysis: female? Age? virgin, pregnant, lactating? which salivary gland was taken? submandibular, parotid, sublingual? How was SG tissue prepared for ChIP-seq and RNA-seq?

Providing an overview (table) at what stages tissues were harvested and which analysis were done (ChIP-seq, RNA-seq, etc) and for which mutants would be helpful to the reader.

NF1- antibody information is not provided.

identification of complex mammary loci section

Mammary specific implies genes that are specifically (only) expressed in the mammary gland. What is described here are pregnancy/lactation induced genes. That some of these are most likely specifically expressed in the mammary gland can be assumed based on the specialized nature of the mammary gland function but this is not a way to determine tissue specificity.

Reference to "the Hi-C data" in the identification of complex mammary loci section? how was this data generated, what are the results? If previously generated provide reference otherwise include generation of this data and results here.

New and reused data

Provide more clarity on the data that was used from previous studies (provide references) and which data was generated in this study, by provide a supplementary summary table indicating which data sets from each accession number were used and associated (reference publication(s))

>> Hi-C and 4C seq data for WT and mutant mice at L1 and ChIP-seq data for WT and mutant mice were uploaded to GSE127144 (ChIP-seq in GSE127139, RNA-seq in GSE127140).<<
Where is this reported in the manuscript??

A table with sequence and alignment statistic should be provided.

How were differences between WT and MUT Samples in ChIP-seq determined? How consisted are the replicates?

I happened to notice, in the RNA-seq data table, differential & high gene expression of certain muscle specific genes indicating contamination with muscle tissue of some samples (which can happen when dissecting mammary glands) this could affect differential gene expression analysis of genes expressed at low to moderate levels. How do all the RNA-seq data cluster in a PCA plot? What are the confounding factors?

Figures and tables

As mentioned above, please indicate clearly what physiological stage the mice were in for each figure and table (Virgin, p6, p18, L1 or L10)

Please use the same way of displaying for RNA-seq based data, and q-PCR derived data.

Figure 2c: why use relative fold of normalized read count here and every where else normalized read counts?

Figure 4b & 5b (and Supl fig6a): depict qPCR results for a Csn gene in WT vs enhancer mutant mammary gland, why is the way this is displayed different? Csn3 Relative gene expression Log10 fold and the Csn1s2 as relative Csn1s1 expression normalized to 1 for WT. Please depict the same way.

Supp fig 3: show the whole Locus and coverage plots for each of the promoter and enhancer regions to illustrate the changes.

Supp fig 4: what happens to Stat5 and K27Ac at Csn1s1 and Csn2 for these deletions? Please show.

Supp fig 6 indicate SE in figure; it would be better to show of deleting Odam on whole locus.

Supplemental Table 8: make sure motifs are indicated in all enhancer regions and suggest also highlighting the GR or GR-half sites and deleted motifs in Scn1s1 E1 and E2

Supp Table 4 & 5 legends: "... each replicate at salivary tissue.." delete "at" (but list what physiological stage the mice where in when tissue was collected)

Reviewer #3 (Remarks to the Author):

In the manuscript "Cell-specific and shared enhancers control a high-density multi-gene locus active in mammary and salivary glands" submitted by Lee and colleagues to Nature Communications, the authors set out to dissect the many different enhancer interactions within a 330kb region that is active in the mammary gland and in the salivary gland. The authors generate many different mouse models with small and big deletions of individual and combinations of enhancer elements and analyze the expression of the genes within the locus mostly in the mammary gland but also in the salivary gland. Here, the main conclusions are that a super enhancer is active in both tissues but has very different sets of target genes. In addition, this main enhancer is not the only one active, there are a few more enhancer elements that are active in this locus that are specific for single genes but then are in turn also affected either by the Super enhancer or affect each other.

Overall, this is a really interesting paper that addresses an important question within the community: how are individual enhancer working together and how can one enhancer have different target genes. The authors underwent a tour de force of generating new and interesting mouse lines, some of the conclusions are not supported by their data in the current form and warrant additional careful examination.

Main concerns:

- Quantifications are missing for the ChIP-seq. Most of the conclusions are only shown in the genome browser views and lack any kind of quantitative comparison. For some conclusions it is necessary to include here some more stringent comparisons. For example, the authors state that for each ChIP two replicates were prepared. However, in all browser views only one replicate is shown. Also, not affected regions that are also differentially regulated during development or lactation could be used as a comparison to show that in the rest of the mammary tissue the initiation of differentiation occurred as expected. Here for example the Wap locus could be used. Then many of the ChIP-seq tracks show very different scales, how was that chosen? Are the ChIP-seq signals that different from each other? For example in Figure 4 Stat5 ChIP-seq varies between 100, 170 and 10,

similarly Pol II seems to be more enriched in the delate E1 cell line compared to the WT? Here at least two different loci should be included and shown and maybe used for a better quantification. This is a problem throughout every figure where ChIP-seq is shown and should be addressed in all instances. This is particular important at all points where the authors claim that one enhancer affects other elements.

- One of the main claims of the manuscript is that the same super-enhancer activates a different set of genes depending on the tissue (see the title of the paper). This is for example depicted in the summary figure 6, where supposedly the same TF bind to the SE and then activate different genes. However, that was not shown in this paper and cannot be claimed. The whole 10kb region is deleted, there could be additional elements that are active in the salivary gland that are not active in the mammary gland and vice versa. If indeed the exact same enhancers are needed then the deletion of the individual constituents should also be analyzed in the salivary gland. In addition, a better characterization of the individual components in the salivary gland is needed for example through ATAC-seq. At this point, the authors can only claim that within the same 10kb region, there is activity that is needed in both cell types. However, whether those are the same individual elements is not shown. The concept of a super enhancer remains controversial, and it is unclear whether they really are different from just a bunch of enhancers that are working together. I would suggest to the authors to refrain from claiming that this 10kb of DNA only functions as one entity but that it might contain different constituents that are active in different tissues.

Minor concerns:

Regarding Supplemental Figure 2

- In the text it says that Supplemental Figure 2 shows that only 12 out of the 20 Stat5 binding sites do actually have a GAS motif. That is not shown or indicated anywhere in this figure.

- Supplemental Figure 2a: it is unclear what the distance refers to here. Is this the distance from a promoter? The distance from an enhancer or (as I suspect) the distance from either a promoter or an enhancer? Then showing this in the same figure should be indicated better. Maybe in this case using heatmaps surrounding the enhancers and the promoters is a better way of depicting it? Alternatively, using a gene body distribution of Pol II for example would be a better way of showing these data.

Regarding Figure 2b: the information in this figure is not clear: the whole super-enhancer cluster is deleted, therefore there should not be anything in the samples to be mapped to the region shown in Figure 2b? I suggest to use a custom annotation for mapping the reads to the genome, since there should not be a signal in these regions. If the authors want to ensure that the regions is really deleted, then including the Sanger sequencing is a better way to convince a reviewer.

Figure 2d: shows changes across the whole locus. When the SE is lost, then the individual enhancer at the Csn3 locus also loses H3K27ac. Here a better quantification is necessary (see above). A second conclusion here is that Stat5 binding at the proximal enhancer to Csn1sb2 is lost. Is this Stat5 binding happening at region with a GAS motif?

Similar: Supplemental Figure 4: loss of the E2 leads to loss of Stat5 binding. Is that again a real GAS binding event or is it a secondary effect?

Regarding Supplemental Figure 3:

- B shows the loss of H3K27ac at enhancers. Here it needs to be clarified whether this is at all enhancers or just a certain subset. Also: a control experiment needs to be included looking at enhancers that are not affected, for example from the Wap locus or the Cish locus as presented in this figure.

Regarding Supplemental Figure 5:

- It is very interesting that the same SE is also active early in pregnancy and that multiple genes are downregulated at the analyzed time points. The authors claim that loss of the SE affects the initiation of casein enhancers. From the figure shown this is really hard to judge: more rigorous quantifications and normalizations are needed, comparison to a non-affected locus should be included that is specifically expressed at this point of development. Which enhancers are affected in the opinion of the authors?

Line 131/132 and discussion: the comment about the distance dependency about the SE would be good to be discussed in the light of the two recent publications from the Giorgetti lab (Zuin et al, Nature 2022) and de Laat group (Rinzema et al., NSMB, 2022).

Line 149/150: Interactions between the Csn3 enhancers and the SE had been confirmed by 3C, further supporting their crosstalk. How does this crosstalk look like? Loss of the E2 enhancer, so the promoter proximal enhancer, abrogates most transcription. Does that now mean that the SE is not needed for the activation of the Csn3 gene? Or that the E2 element is needed for the activation of both E1 and the SE on the gene?

Line 189: "additive activity in the alpha globin locus". Upon reexamination of the alpha globin locus in a recent Biorxiv paper, Higgs and colleagues have shown that it is not just simple additivity that works at the locus, but that some elements are needed for the locus to fully activate even though they do not have their own activity (Blayney et al, BioRxiv, 2022).

We wish to thank the reviewers for the time they spent carefully reading our manuscript and their critiques and suggestions, which we implemented.

As requested by reviewer #2, and as discussed with the Editor, Dr. Mononen, we have now included the results from our companion study (**A cytokine-responsive promoter is required for distal enhancer function mediating the hundreds-fold increase in milk protein gene expression during lactation**). We also included newly generated data analyzing the global DNA methylation status (Methyl-seq) in lactating mammary tissue, with emphasis of all genes in the casein locus (Supplementary Figure 6). As requested, we also included additional data demonstrating the significance of enhancers and promoters in late lactation.

We believe that the integration of all data from the two studies provides comprehensive insight into the regulation of the complex casein locus in mammary and salivary tissues. The study now includes results from the four-module super-enhancer, the local enhancers of *Csn1s1*, *Csn2* and *Csn3* and the *Csn2* promoter. In total, we investigated the biological function of 12 regulatory elements, individually and in relevant combinations, for which we generated and analyzed 21 new lines of mutant mice. For this study we generated and analyzed a total of 167 new NGS data sets (124 ChIP-seq, 38 RNA-seq and 5 methyl-seq) that were now deposited in GEO ([GSE231441](https://www.ncbi.nlm.nih.gov/geo/query/acc.cgi?acc=GSE231441), token: uhizwsympjevdsd). For a summary of these data, see Supplementary Table 13.

Reviewer #1 (Remarks to the Author):

Summary:

In the manuscript titled “Cell-specific and shared enhancers control a high-density multi-gene locus active in mammary and salivary glands”, Lee et al. have investigated the role of enhancers and super-enhancers (SE) in the regulation of genes present in a specific locus that has relevant roles in both mammary- and salivary-gland tissues. The most interesting aspect of this study is the choice of locus they have chosen to study given both its complexity – it contains 8 genes and 20 candidate regulatory elements – and its mutually shared as well as exclusive activity in the 2 distinct tissues. The overall approach of employing experimental mouse genetics for this study is also sound and frequently yields intriguing results. However, significant concerns regarding (a) the rationale and (b) interpretation of those results temper the conclusions claimed by the authors in this manuscript. Additionally, several minor transgressions in the writing also accumulate to form a major concern regarding the inferences supporting the conclusions of the study.

Major concerns:

1. The authors need to be consistent in the organization and presentation of related data of their figures. For example, in different instances, the authors display comparative RNA-Seq results as “relative fold of normalized read counts” and “relative gene expression (fold)”, “normalized read counts (log10)” instead of using one standard metric for such comparisons. Such choices raise additional questions regarding the methods used to make these comparisons, that are neither addressed in the figure legends nor in the methods section.

Response

We thank the reviewer for pointing this out. We have used both RNA-seq data (normalized read counts) and relative gene expression using qRT-PCR data. Except for figure 1, we used relative gene expression in RNA-seq and qRT-PCR data. Each figure legend describes exactly the experimental approach used.

2. Genome tracks do not have comparative scales. Since the height of the peaks depend on sequencing depth and transcript abundance, we understand (with some reservations) the choice of using different scales to simply compare the presence and absence of peaks in the WT and the experimental (deletions) track. However, this approach has a major pitfall when comparing the different deletions. For example (but applies to almost every figure), in supplementary figure 4, STAT5 binding in $\Delta E2/E4$ is significantly increased at E1 (160 vs 75 on y-axis) – does it indicate compensatory binding at E1 when E2 and E4 are not available? What does it mean for the expression and regulation of the genes under the control of the enhancer? The author must consider these questions, as they're fundamental to the understanding of gene regulation at this locus. Additionally, the H3K27ac peaks at E1 for $\Delta E1$ (absent) and $\Delta E2$ (present) could be quite comparable since the low peaks in $\Delta E1$ are scaled to 60 and in $\Delta E2$ are scaled to 10. The authors might be aiming to compare the relative enrichment of peaks in E1 and E4, but this approach is ultimately biased and we must recommend extreme caution when making inferences from such results.

Response

We thank the reviewer for pointing out these important issues. The 'deepTools' contain useful modules to process the mapped reads data for multiple quality checks, creating normalized coverage files in standard bedGraph file formats. We used this program to normalize ChIP-seq peaks.

The peak heights depend on efficiencies in nuclear extraction, DNA shearing and immunoprecipitation, and the sequencing depth, which can be different between experiments. To clearly visualize this, we had included ChIP-seq images of a STAT5 target gene located on another chromosome that was not affected by the enhancer deletions. In this case we had added the Cish gene, and it is clear from the image that the STAT5 peaks are increased in the E2/E4 mutations. Thus, the increased STAT5 E1 peaks in the SE are not the result of the E2/4 deletions but rather a secondary consequence of the sequencing depth. We addressed these issues in a new paragraph in the Material and Methods section (page 16-17, lines 490-505). Also, deletion of E4 did not yield expression changes in any of the casein genes as shown in the figure. The same explanation applies to the H3K27ac marks.

This purpose of this supplementary figure was to demonstrate the biological consequences of deleting individual enhancer modules and the answer is clear, E1 and E2 have enhancer activity and their combined deletion silences the SE to the same extent as the deletion of the entire SE (E1-4).

3. In several figures involving genome tracks, why do we observe any peaks in the locus that are specifically deleted in the ChIP-Seq samples? This is likely background, but raises the question of why the authors haven't normalized the samples with, for example, an IgG control?

Response

We wanted to confirm of mutations using ChIP-seq data and show its activity in controlling genes by comparing between wild type and mutant mice. We couldn't see any sharp peaks in IgG control data.

4. The authors provide a very skewed perspective on the alleged evolutionary changes observed in the locus – from their writing, they seem to indicate that the high-density multi-gene locus evolved with the goal of somehow enhancing the regulatory capabilities of the various elements that resulted from events such as gene duplication etc. and is not merely a characteristic that is more adaptive in nature. To adequately support their claims, the authors must also employ other species in their experimental design, even if to merely address it from a genomic perspective.

Response

We did not intend to imply that the locus evolved with the goal to enhance regulatory capabilities. It was our intention to point out that a simple ancient locus encoding a gene expressed in salivary tissue expanded and additional regulatory elements evolved in the mammalian lineage that facilitated expression in mammary tissue. We now added a figure depicting the structure of locus in other extant mammals. As can be seen from the gene arrangement, this is a very active locus.

Minor concerns:

5. The authors have used fold-change (log-normalized or otherwise) to show relative mRNA enrichment in comparisons of different samples, then they should be consistent in their text by reporting relative expression changes in fold-changes and not %-ages.

Response

We used relative gene expression for RNA-seq and qRT-PCR data, except for figure 1 (normalized read counts from RNA-seq). We ensured consistency throughout the figures.

6. Through the writing, the authors must attempt to make the paper more accessible to a broader scientific audience. The writing is rife with jargons that are specific to the niche community (for example, p6 can mean “pregnancy day 6” as well as “post-natal day 6” to different group of readers) and the authors must make the effort to define such specific terms in the main text or the methods.

Response

We thank the reviewer and agree that the writing had to be clarified. We now avoid any jargon so people outside the mammary field can have a better understanding of our study. We also refrained from using abbreviations such as p6 as much as possible and rather spelled them out (e.g., day 6 of pregnancy).

7. The authors must improve the language of the manuscript more generally. Each paragraph must have a clear rationale and proper conclusion to ensure flow of logic, which is largely missing from the manuscript making it a generally difficult read. Specific examples of concerning language include (but are not limited to):

- a. Page 1, line 30: "...success of milk and saliva" – given that success refers to the outcome of a process, which process involving milk and saliva are the authors referring to, should be made clearer.
- b. Page 1, line 30-31: "...mineralized tissues, like mammary and salivary glands" – examples of mineralized tissues include nails and teeth. Mammary glands and salivary glands are not mineralized tissues.
- c. Page 2, line 61: "...has all the hallmarks of a super-enhancer" – which specific hallmarks are the authors referring to? The increased H3K27ac peaks?
- d. Page 3, line 63: "chromatin structures" – the authors have not assessed structures, but protein occupancy.
- e. Page 4, line 94: "to understand... we conducted additional ChIP-seq experiments" – but with what changes in conditions?

Response

The manuscript seen by the reviewers was originally submitted as a short letter to Nature Genetics, and it was subsequently transferred without reformatting to Nature Communications. We have now expanded the manuscript, added the data from the Csn2 enhancer-promoter study and Methyl-seq data that identify receptive promoters in the casein locus. We now formatted the manuscript in the traditional way and made it more accessible to a wider audience.

- A. *Milk was the key to the success of mammals, and we stated this in lines 29-31 (The transformation of proto-lacteal fluid rich in calcium binding proteins into nutritious milk was vital for the sustained success of mammals).*
- B. *We corrected this statement.*
- C. *Hallmarks are defined by the Rose algorithm, the presence of three or more enhancer modules within a region of ~ 10 kbp. We added this to the text (line 105).*
- D. *We agree and changed the sentence to 'Integration of active histone marks, transcription factor binding, and gene expression data suggest that the mammary-*

salivary locus harbors the highest density of candidate enhancers and highly regulated genes among all multi-gene loci in mammary tissue (Supplementary Fig. 1b-d, Supplementary Table 2)' (lines 107-110)

E. The 'additional' referred to ChIP-seq experiments in the super-enhancer knock-out mice. We now state: 'To understand whether the SE is required for the establishment of gene-specific regulatory elements, such as enhancers, we conducted ChIP-seq experiments in tissue lacking the SE.' (lines 147-150).

8. Most of the genome tracks do not have consistent labeling, which results in significant confusion on part of the reader.

Response

We now used consistent labels in all figures.

Reviewer #2 (Remarks to the Author):

In this manuscript the authors describe an extensive study of the regulatory landscape of the casein gene locus which contains genes expressed in epithelial cells, 5 casein genes expressed predominantly and at high levels in the lactating mammary gland epithelium and 3 genes expressed predominantly in the salivary gland epithelium among other tissues, one of which is also detected in the lactating mammary gland. This is a real tour the force with the sheer number of deletion-mutant mice analyzed for different aspects of transcription factor binding, chromatin marks and expression. It gives fascinating insights in the complexity of the regulation of a large multi gene locus predominantly expressed in two different epithelial tissues. It makes clear that both larger Super-enhancers and smaller locally acting enhancers and promoters are important to achieve full functionality, and that some level of redundancy might exist to safeguards a function as important for species survival as lactation. The aspect of temporal regulation in the mammary gland is however not well addressed and could be worked out better especially within the context of previous work by these authors and others and the reported findings in this manuscript related to effects in early pregnancy.

Response

We thank the reviewer for the positive assessment of our work. We have now included additional data covering gene expression in mammary tissue at days 1 and 10 of lactation, day 6 of pregnancy and in virgin (non-parous) mice. As stated in the text, deletion of the super-enhancer and one local Csn3 enhancer resulted in lactation failure and the respective gene expression studies were therefore conducted at day 18 of pregnancy, just prior to parturition. A summary of all mutant mice and the time points chosen for the analyses is shown in Supplementary Table 14.

Conceptually, I Strongly suggest including the Csn2-enhancer data reported in the accompanying paper to provide a complete picture especially as this is part of the locus and just as all other casein genes Prl-Jak/stat regulated. I do not think much is gained by reporting this separately and not in the current form.

Response

We fully agree with the reviewer and included all data from the companion manuscript. We added the data covering five regulatory elements (three enhancers and two promoter elements) of the Csn2 gene for which we generated nine mouse lines. These data are presented in Fig. 7-8, Supplementary Fig. 8-9 and Supplementary Tables 9-11.

My main problem is that in general, the manuscript fails to contextualize the reported findings with previous work reported by others on regulatory elements, chromatin organization, epigenetic modifications and expression at different physiological time points in the mammary gland. Most of the regulatory regions identified here, are conserved and have been identified before by others in mouse and other mammals and have been studied in the context of chromatin organization and chromatin interactions, gene regulation, and transcription factor binding in the mammary gland and MEC at

different physiological states. E.g. Csn1s1 E2 (ECR3 mouse/cow-development/differentiation, Cow -mastitis and involution) Csn2-E1 = BCE (conserved) identified in human, cow and mouse, studied in vitro and in vivo (although this region is not covered in this manuscript but the accompanying manuscript), Mouse ECR19 = SE-E1).

Response

We thank the reviewer for pointing out the inclusion of data reported in the literature. We have contextualized our findings with previous work on this locus. We have presented our work in the context of research published by other groups, including Rosen and colleagues, dosSantos and colleagues and others (references: 5, 14, 15, 20, 21, 50, 51, 52, 54).

The authors fail to mention that the SE mutant mice show lactation failure (mentioned in a previous paper by the authors) even though this can be expected when Csn3 expression is virtually wiped out (DOI: 10.1073/pnas.0601611103) (Kumar, Csn3 deletion), this is not addressed in the manuscript, nor is the expected consequence of the different enhancer deletions and mutations based on results of Csn1s1 and csn2 KO mice discussed. Csn1s1, Csn2 and Csn3 knocked-out mice have been reported with different effects on milk production. This probably explains why expression is analyzed in P18 tissues, this needs to be made clear and discussed.

Response

We have now discussed our findings in the context of our previous study (reference: 17) in which we deleted the four CTCF sites in the casein locus (lines 358-362). Our previous study demonstrated a moderate contribution of the CTCF sites located within the SE and only expression of the Csn1s1 gene was slightly induced in mice from which the SE-CTCF site was deleted. We did not analyze gene expression in salivary glands and the mice are no longer available.

It also would be helpful to discuss the finding in the context of previously reported CTCF work by the authors. What is the role of CTCF in the mammary gland vs SG? And how do the deletions described here correlate to CTCF site/binding and how could this interact/affect CTCF binding/function. At least discuss this.

Response

We were unable to identify publications reporting deletions of CTCF sites in the mouse genome followed by studies in salivary gland tissue. The deletion of the entire SE also included the two CTCF sites. If these CTCF normally block the sphere of influence of enhancers linked to Csn1s1, Csn2 and Csn1s2b then we would see an increased expression of Odam, Fdcsp and Csn3, which we did not. These three genes are strictly under the SE control and the other enhancers within the locus could not compensate for its loss.

As this work spans assessment of several regulatory elements in the casein locus it would be very helpful to put this in perspective to previous work by others and enables readers to understand the state of the field, and the confirmation of previous work and the new contributions as reported in this study.

Response

We now report previous research in detail and place it in context of our findings. For example, STAT5 binding to the candidate enhancer E1 structure in the Csn2 locus had been identified in HC11 cells by the Rosen group (references: 50-51) and we now deleted this site, which by itself seems to be not necessary (lines 417-420). The distal Csn2 candidate enhancers had not been reported prior to our study. Our study provides and significant information as it uses experimental genetics to address the functions of regulatory elements.

To aid in reader's understanding add details on which time points (p6, p18, L1, L10) are being analyzed and reported on (in text, figures and figure legends) and why certain time points are selected.

Response

We now added Supplementary Table 14 detailing the time points analyzed for each mutant. In general, we analyzed tissue from virgin mice, day 6 and 18 of pregnancy and days 1 and 10 of lactation. For those mouse lines that failed to lactate due to impaired Csn3 expression we did not analyze tissue post-delivery.

Temporal aspects of the regulation should be addressed more thoroughly, as it has been reported that regions encompassed in the SE (SE-E1) as well as Csn1s1-E2 show open chromatin (based on histone marks and DNA hypomethylation) in virgin MEC and virgin MEC subpopulations (Rijnkels, 2013 and DosSantos, 2015)

Response

We carefully read the literature and integrated the data from the Rosen group (references: 5, 14, 15, 20, 50, 51, 54) and DosSantos (reference: 21). These data are shown in Supplementary Figure 5 (H3K4me2 data from Rijnkels and colleagues) and Supplementary Figure 6d-f (DNA methylation data from DosSantos). We reanalyzed the DosSantos data, and we are uncertain about the open chromatin at Csn1s1-E2 in virgin mice. However, our newly generated DNA methylation data from lactating mammary tissue demonstrates hypomethylated promoter regions in the casein genes and most profound in the Csn1s1 gene (lines 200-209).

9. What is the effect of deleting the SE, Csn1s1-E1 or E2, or Csn3 E1, E2 or P on virgin casein and other locus gene expression as well as expression at the different developmental stages including L1 and L10. The locus undergoes gradual changes in

chromatin organization with functional differentiation and milk protein production how is that reflected in the effect of the different enhancer/ promoter mutants?

Response

The focus of the work was the induction of casein genes during pregnancy with days 1 and 10 of lactation as endpoints. Yes, we agree that data from virgin tissue would enhance the value of the work and we added such data where applicable. H3KK27ac and Pol II ChIP-seq data from virgins were added in Supplementary Fig. 5. We also included H3K4me2 data from the Rijnkels study (reference: 20). Data from virgin mice carrying Csn2 mutations were added in Supplementary Figure 8, and they demonstrate that the identified promoter and enhancer elements are not required for basal (non-pregnancy induced) expression. We also included L10 data from mice carrying the Csn2 promoter and enhancer mutations (Supplementary Fig. 8). For the SE deletion we added data from day 6 of pregnancy (Fig. 2d-e). As mentioned earlier deletion of the super-enhancer and Csn3-E1 resulted in lactation failure, and we could reliably analyze only timepoints prior to parturition. All data points collected and analyzed are depicted in Supplementary Table 14. The results from key mutations are summarized in the heatmap plot in Fig. 9a.

10. It would be helpful to show Chromatin enhancer markers/ TF binding of the different regulatory elements in the whole locus for the different deletions, as well as expression of all genes in the locus, preferably at the different time points. Currently only subsets of the locus are shown in several cases.

Response

Now, we edited all figures and present the entire casein locus and the Cish locus as a control from all mutant mice.

11. Showing the coverage trace-difference changes in binding/enrichment using coverage profiles for the different regulatory regions for P6, P18, L1, L10 and WT vs Mut, similar to the coverage plots in Lee et al 2018, will help to clearly visualize changes. The current aggregate coverage plots do not visualize this and the differences in coverage are hard to see in some cases in genome browser view images.

Response

We show coverage plots for enhancers and promoters in the locus in Supplementary Fig. 2 and for WT and Δ SE mutant mouse who show difference in the entire locus in Supplementary Fig. 3.

12. Based on the binding of the TF analyzed and timing and the effect of deleting certain enhancers. Can you hypothesize which factor(s) aid in opening chromatin in the different enhancers and the timing of it?

Response

This is a very good question, and we wish we could answer it. As the reviewer certainly knows, the presence of TF binding motifs does not tell the whole story. Future experiments would need to focus on very early stages of mammary development, even embryonic stages and use more sensitive assays to identify regulatory regions. There is, of course, the issue of the amount of tissue needed for a ChIP-seq experiments and the quality of antibodies. In general, we need mammary tissue from 5-10 mature virgin mice for one ChIP-seq experiment.

13. other factors besides STAT5, NF1, GR have been shown to play a role in regulation of genes in this locus (e.g., ELF5, CEBP/B, even PR and YY1) I know not all of these can be studied but some mention is warranted.

Response

We have now included publications on ELF5 and the PR in the discussion (lines 367-370).

14. In figure 1 and sup figure 1 instead of showing p6 vs L1 and virgin vs L10 for WT and then showing several p18 WT vs MUT RNA-seq data sets, can you show expression changes in the casein locus genes from Virgin through L10 in one figure?

Response

We added bar graphs for RNA-seq from mammary tissue at virgin, pregnancy day 6, lactation day 1 and 10 in Fig. 1c.

Other comments:

15. Unconventional manuscript structure: Introductory paragraph, main text, materials and methods. Instead of Abstract, introduction, results, discussion materials and methods.

Response

The original version of the manuscript had been written as a short report for Nature Genetics and it was then transferred to Nature Communications. We now reformatted the manuscript and the revised version has all sections.

16. Use of mammary specific vs preferential expression, be consistent. I suggest preferential expression.

Response

We agree and we now refer to mammary preferential expression.

17. Same nomenclature should be adopted throughout the manuscript (including Materials and Methods) for the enhancers/ promoters that includes the gene name for

which the promoter or different enhancers are mutated/ investigated, there are too many E1's and E2's.

Response

We fully agree with the reviewer and added gene names in mutant names. For example, 'Csn1s1-E1'.

18. In the "introductory paragraph" (abstract?)

It might be useful to mention all genes present in the locus by name, especially because two are mentioned subsequently.

Response

We mentioned in all genes in the abstract.

19. Line 21/22: "the other three caseins" only one of the five has been mentioned >> revise as appropriate by mentioning csn1s2b (I assume) or rephrasing. Also make clear when in the mammary gland the 98% reduction takes place.

Response

We edited the abstract.

20. Line 30/31: salivary gland and mammary gland are not mineralized tissues; they are secretory epithelial tissues and incorporate calcium in their secretion. And the genes in this region encode secretory calcium binding phospho-proteins, contributing in the case of the Odam gene to dental mineralization. ref 2 and 3 are appropriate, other references by the same authors specifically addressing the duplication giving rise to the casein locus are more appropriate.

Response

We corrected the sentences with references.

21. Line 190/191 please provide appropriate references.

Response

We added references 41 and 42.

22. Prr27 is not reported on, why?

Response

Prr27 was not detected by qRT-PCR and RNA-seq and must be expressed at very low levels.

23. For the stich analysis, please share the final list of identified complex loci and the genes they contain in a supplementary table.

Response

This information is now reported in Supplementary Table 2.

24. It seems that the identification of regulatory elements was focused only on those that contain STAT5 binding.

Response

Yes, we focused on regions that were bound by STAT5, GR and NFIB and coincided with activating histone marks. These are also the most highly expressed TFs in mammary tissue. We did not observe extended H3K27ac enriched areas that were not recognized by STAT5. Naturally, future experiments could focus on additional TFs for which ChIP-seq validated antibodies are available.

25. If a piece of DNA is deleted, as is the case in most of the enhancer mutants reported on here, it is expected that you do not detect binding of transcription factors or presence of histone marks, as the DNA is not there to bind or even to be sequenced. The lack of detection of such binding confirms the deletion but nothing else, so pointing out the lack of binding is a bit fraught, especially for TFs binding. It might be worth pointing out the loss of histone marks in the surrounding DNA as this implies that the deleted DNA might play a role in setting up the chromatin mark.

Response

Now, we presented ChIP-seq data to the entire casein locus in all figures.

26. Individual SE enhancer mutations

What happens to the Binding of STAT5, GR, NF1 and chromatin marks at *Csn1s1*, *Csn2*, *Csn1s2a* upon deletion of the individual SE enhancers dE1-4?

Response

*TF (STAT5, GR, NFIB) binding to the *Csn1s1*, *Csn2* and *Csn1s2a* genes was not altered in in was not affected in *Csn-ΔE1/2* and *ΔSE* mammary tissue (Fig. 2c and Supplementary Fig. 4). Some reduction of H3K27ac was observed at all casein genes in *ΔSE* tissue (Fig. 1c). Pol II loading was reduced only in the *Csn3* locus.*

27. Do you see similar changes in expression as for the whole SE deletion at earlier time

points?

Response

We now analyzed expression data at day six of pregnancy (Fig. 2d-e) and found all Casein genes to be affected. Thus, the SE has a more extended function in early pregnancy (Lines 154-163).

28. What happens in the salivary gland with the individual dE1-4 deletions? Can you predict other TF in these regions that could be active in SG?

Response

Deletion of the entire SE (Δ SE) resulted in the loss of Odam, Fdcsp and Csn3 gene expression (Fig. 3). We have not investigated the consequence of individual deletions and these mice are not available any longer. The pandemic caused NIH to shut down in the spring of 2020 and were mandated to cull our mouse colony. We were permitted to keep breeding pairs of the most important lines.

Key TFs controlling mammary function, such as ELF5, are dispensable in the salivary gland (references: 7, 43). We mentioned this in the Discussion part.

29. Does the SE deletion (or Odam deletion) affect expression of salivary and dental epithelial genes located more distal to the casein locus (such as smr, prol genes or Ambn, Enam, Amel?)

Response

The expression of other salivary genes (Smr3a and Smr2) was not changed and dental genes (Amtn, Ambn, Enam) are expressed very low level or non-expressed in salivary tissue of Δ SE mouse compared to WT (Supplementary Table 6).

30. Csn1s1-E1 and -E2 mutations:

The deletion of the GR motif in E1 also leads to loss of GR at the promoter (this is not mentioned), but this does not appear to change expression levels. With respect to the E2 mutation, although it is not clear at what stage of lactation the expression of Csn1s1 was analyzed (L1,L10?) a 65% reduction in expression is significant in the context of milk production/ nutritional value, if that translates to a reduction in protein content of the milk. If this is at L1 is this more profound at L10 or less so? What happens to expression of the other Csn genes and non-Csn genes in the locus? It is worth noting that Csn1s1-E2 is the same as ECR3 reported by Rijnkels et al (2013), which has an open chromatin conformation in virgin MEC just like SE-E1 (Rijnkels 2013, DosSantos 2015). As the SE deletion shows more profound effects on gene expression at P6 or earlier what happens to the expression of genes in the locus upon mutation of Csn1s1-E2?

In the Csn1s1-E2mutant it appears that STAT5 binding is only lost at E1 (and reduced at the promoter), while GR binding is lost at E1 and the promoter, which could lead to

impaired regulation of *Csn1s1* expression, it should be pointed out in the text. As it is unclear at what stage the ChIP was done, qPCR is lactation (L1 or L10?), does this change at L10? Or at earlier time points? Does the deletion affect ELF5 binding site too?

Response

*We specified the time point of data set (day 1 of lactation, L1) in the manuscript. The mutant mice were analyzed at L1 and L10 and showed the same results. Also, mice carrying individual mutations in SE modules did not have any over problems with lactations, the pups developed just fine. This is evidence that E1 or E2 alone can drive sufficient *Csn3* expression.*

*We analyzed methyl-seq data (DosSantos) and ChIP-seq data (Rijnkels) from the respective publications and added them in Supplementary Fig. 5-6, together with our virgin ChIP-seq and bisulfite-seq data. Although there are some H3K4me2 marks at E1 of the SE and possibly at *Csn1s1*, there is no active H3K27ac and Pol II coverage in virgin tissues. Therefore, we suggest the regulatory elements in the casein locus are largely established by pregnancy hormone prolactin.*

As the reviewer suggested, we added the findings from ChIP-seq data in the results part (lines 173-177). Since we deleted the entire E2 region, ELF5 binding site is not in the region of mutant mice.

General questions related to the generation of deletion mutants via CRISPR/Cas9.

31. Can you rule out off target effects? As several of the gRNA target regulatory elements with e.g. STAT5 sites and many milk protein genes are regulated by the same set of transcription factors.

Response

When we design sgRNAs, we checked the potential for off-target effect and excluded those with a high propensity for off-target effect in the sgRNA list. Also, since we targeted enhancer regions, but not only STAT5 binding site (TTCNNGAA), it is not possible sgRNAs targets multiple enhancers. Finally, we validated the deletion through Sanger sequencing (Supplementary Table 13) and ChIP-seq.

32. What happens to chromatin marks and TF binding at other milk protein genes and milk fat synthesis related genes, and what happens to their expression?

Response

*Our lab has analyzed ChIP-seq data for TFs and histone markers in genome wide and reported their binding feature on all mammary genes and the *Wap* gene (PMID: 30285185, 27376239, 24678731, 24277936). Prolactin induces their binding on milk protein genes in mammary tissue during pregnancy and lactation. The enhancer and promoter deletions investigated in this study did not affect other milk protein genes. They also did not influence expression of genes outside the casein locus that is flanked by CTCF sites as shown in our earlier study (reference: 17).*

33. The SE deletions and Odam deletions are large deletions, 8-10KB, how do they affect 3D organization of the locus? and what is the effect of moving certain enhancers and promoters closer together?

Response

In our previous study (reference: 23), we showed interactions between the SE and enhancers of Csn1s2b and Csn3 genes. They should disappear in Δ SE mutant. In our previous study on the Wap gene (PMID: 30285185), we demonstrated that moving the Wap-SE closer to a neighboring gene, that is normally expressed at very low levels in mammary tissue, can slightly induce its expression.

However, when we deleted the Odam gene and moved the SE closely to the Fdcsp gene, we did not detect any expression in mammary tissue, suggesting that the Fdcsp promoter is not receptive (lines 194-196) Our newly added DNA methylation data demonstrate high methylation surrounding the Fdcsp TSS (Supplementary Figure 6, lines: 204-206). However, Csn3 expression in mammary tissue doubled.

Materials and Methods:

34. Provide details for salivary gland analysis: female? Age? virgin, pregnant, lactating? which salivary gland was taken? submandibular, parotid, sublingual? How was SG tissue prepared for CHIP-seq and RNA-seq?

Response

We added this information about how mice were used in all assays in the method part. Mammary and salivary tissues were processed in the same way for CHIP-seq and RNA-seq and they were collected and stored at -80°C until performing assay.

35. Providing an overview (table) at what stages tissues were harvested and which analysis were done (CHIP-seq, RNA-seq, etc) and for which mutants would be helpful to the reader.

Response

We summarized the information in Supplementary Table 14.

36. NF1- antibody information is not provided.

Response

We added its information in the Method part.

37. Mammary specific implies genes that are specifically (only) expressed in the mammary gland. What is described here are pregnancy/lactation induced genes. That

some of these are most likely specifically expressed in the mammary gland can be assumed based on the specialized nature of the mammary gland function but this is not a way to determine tissue specificity.

Response

Well, the reviewer is correct and there is (probably) no absolute cell specificity. Sequencing RNA deep enough frequently detects low level expression in other tissue. Since GAS motifs are recognized by STATs, it is well possible that casein genes are expressed at (very) low levels in some immune cells.

38. Reference to “the Hi-C data” in the identification of complex mammary loci section? how was this data generated, what are the results? If previously generated provide reference otherwise include generation of this data and results here.

Response

We didn't use Hi-C data in this manuscript and deleted it in the Method part.

New and reused data

39. Provide more clarity on the data that was used from previous studies (provide references) and which data was generated in this study, by provide a supplementary summary table indicating which data sets from each accession number were used and associated (reference publication(s))

Response

We summarized data set used in this manuscript in Supplementary Table 14.

40. >> Hi-C and 4C seq data for WT and mutant mice at L1 and ChIP-seq data for WT and mutant mice were uploaded to GSE127144 (ChIP-seq in GSE127139, RNA-seq in GSE127140).<<

Were is this reported in the manuscript??

Response

We did not use Hi-C data in this manuscript and deleted it in the Method part.

41. A table with sequence and alignment statistic should be provided.

Response

We provided tables for sgRNA sequences and Sanger sequences of mutated regions in Supplementary Tables 12-13.

42. How were differences between WT and MUT Samples in CHIP-seq determined? How consisted are the replicates?

Response

Although sequencing depth and CHIP-seq peak height could be different between replicates, the peak patterns are the same. We added Spearman correlation in Supplementary Table 14.

43. I happened to notice, in the RNA-seq data table, differential & high gene expression of certain muscle specific genes indicating contamination with muscle tissue of some samples (which can happen when dissecting mammary glands) this could affect differential gene expression analysis of genes expressed at low to moderate levels. How do all the RNA-seq data cluster in a PCA plot? What are the confounding factors?

Response

Mammary gland tissue for RNA-seq was collected from the abdominal mammary gland #4 (red circle) after removing lymph node. Although we carefully dissected the gland, there is always the possibility of muscle contamination.

We would like to point out that basal cells in mammary tissue also express muscle-specific genes.

In PCA plots, most data are categorized by mutant lines and stages.

Figures and tables

44. As mentioned above, please indicate clearly what physiological stage the mice were in for each figure and table (Virgin, p6, p18, L1 or L10)

Response

We added clear time points in figures and figure legend.

45. Please use the same way of displaying for RNA-seq based data, and q-PCR derived data.

Response

We used relative gene expression (fold activity) format in all gene expression figures.

46. Figure 2c: why use relative fold of normalized read count here and every where else normalized read counts?

Response

It was a way to present our results clearly. Now, we displayed all expression data in the same way.

47. Figure 4b & 5b (and Supl fig6a): depict qPCR results for a Csn gene in WT vs enhancer mutant mammary gland, why is the way this is displayed different? Csn3 Relative gene expression Log10 fold and the Csn1s2 as relative Csn1s1 expression normalized to 1 for WT. Please depict the same way.

Response

We used \log_{10} values in Fig. 5b because it permitted us to distinguish the smaller values of Csn3-E2-S and Csn3-E2-S/N.

48. Supp fig 3: show the whole Locus and coverage plots for each of the promoter and enhancer regions to illustrate the changes.

Response

We now show the entire locus.

49. Supp fig 4: what happens to Stat5 and K27Ac at Csn1s1 and Csn2 for these deletions? Please show.

Response

TF (STAT5, GR, NFIB) binding to the Csn1s1, Csn2 and Csn1s2a genes was not altered in in was not affected in *Csn- Δ E1/2* and Δ SE mammary tissue (Fig. 2c and Supplementary Fig. 4). Some reduction of H3K27ac was observed at all casein genes in Δ SE tissue (Fig. 1c). Pol II loading was reduced only in the Csn3 locus.

50. Supp fig 6 indicate SE in figure; it would be better to show of deleting Odam on whole locus.

Response

We edited the figure and covered entire locus.

51. Supplemental Table 8: make sure motifs are indicated in all enhancer regions and suggest also highlighting the GR or GR-half sites and deleted motifs in Csn1s1 E1 and E2

Response

After combining the two manuscript we generated a new supplementary table 13 with all the information. The deletions are shown and the TF binding motifs are shown.

52. Supp Table 4 & 5 legends: "... each replicate at salivary tissue.." delete "at" (but list what physiological stage the mice where in when tissue was collected)

Response

We corrected the legend and added information about what age and stage tissues were harvested in the Method part.

Reviewer #3 (Remarks to the Author):

In the manuscript “Cell-specific and shared enhancers control a high-density multi-gene locus active in mammary and salivary glands” submitted by Lee and colleagues to Nature Communications, the authors set out to dissect the many different enhancer interactions within a 330kb region that is active in the mammary gland and in the salivary gland. The authors generate many different mouse models with small and big deletions of individual and combinations of enhancer elements and analyze the expression of the genes within the locus mostly in the mammary gland but also in the salivary gland. Here, the main conclusions are that a super enhancer is active in both tissues but has very different sets of target genes. In addition, this main enhancer is not the only one active, there are a few more enhancer elements that are active in this locus that are specific for single genes but then are in turn also affected either by the Super enhancer or affect each other.

Overall, this is a really interesting paper that addresses an important question within the community: how individual enhancer are working together and how one enhancer can have different target genes. The authors underwent a tour de force of generating new and interesting mouse lines, some of the conclusions are not supported by their data in the current form and warrant additional careful examination.

Response

We thank the reviewer for the comments. The revised version of the manuscript now includes data from our Csn2 promoter-enhancer study that demonstrates synergism between promoter-bound cytokine response elements and three upstream enhancers. The revised manuscript covers data of 12 regulatory elements, and we analyzed these individually and in relevant combinations in 21 mutant mouse lines.

Main concerns:

53. Quantifications are missing for the ChIP-seq. Most of the conclusions are only shown in the genome browser views and lack any kind of quantitative comparison. For some conclusions it is necessary to include here some more stringent comparisons. For example, the authors state that for each ChIP two replicates were prepared. However, in all browser views only one replicate is shown. Also, not affected regions that are also differentially regulated during development or lactation could be used as a comparison to show that in the rest of the mammary tissue the initiation of differentiation occurred as expected. Here for example the Wap locus could be used. Then many of the ChIP-seq tracks show very different scales, how was that chosen? Are the ChIP-seq signals that different from each other? For example, in Figure 4 Stat5 ChIP-seq varies between 100, 170 and 10, similarly Pol II seems to be more enriched in the delate E1 cell line compared to the WT? Here at least two different loci should be included and shown and maybe used for a better quantification. This is a problem throughout every figure where ChIP-seq is shown and should be addressed in all instances. This is particular important at all points where the authors claim that one enhancer affects other elements.

Response

We thank the reviewer for the constructive comments. We agree that it is important to show ChIP-seq data from loci that were not affected by the enhancer mutations. We show such examples in the Supplementary part. In Fig S3 and S5, we show data from Cish, a gene known to be under STAT5 control.

Each data set (experiment) has different number of total reads which results in different peak heights. In the revision, we used Deeptool to normalize coverage and compare between data sets. As the reviewer suggested, we added Spearman correlation result of replicates (Supplementary Figure 14) and the Wap locus as a ChIP-seq control.

54. One of the main claims of the manuscript is that the same super-enhancer activates a different set of genes depending on the tissue (see the title of the paper). This is for example depicted in the summary figure 6, where supposedly the same TF bind to the SE and then activate different genes. However, that was not shown in this paper and cannot be claimed. The whole 10kb region is deleted, there could be additional elements that are active in the salivary gland that are not active in the mammary gland and vice versa. If indeed the exact same enhancers are needed then the deletion of the individual constituents should also be analyzed in the salivary gland. In addition, a better characterization of the individual components in the salivary gland is needed for example through ATAC-seq. At this point, the authors can only claim that within the same 10kb region, there is activity that is needed in both cell types. However, whether those are the same individual elements is not shown. The concept of a super enhancer remains controversial, and it is unclear whether they really are different from just a bunch of enhancers that are working together. I would suggest to the authors to refrain from claiming that this 10kb of DNA only functions as one entity but that it might contain different constituents that are active in different tissues.

Response

Yes, we fully agree with the reviewer that the activity of the SE in mammary and salivary tissue might require different sets of transcription factors. What we know is that the entire 10kb SE controls gene expression in mammary and salivary tissues. We discussed this in the manuscript and cited a paper (reference 38) that demonstrated different elements with a retinal SE. The reviewer suggested to evaluate salivary gene expression in the individual enhancer deletions. Unfortunately, mice carrying individual deletions are no longer available as the pandemic caused a complete shutdown of the NIH and we had to cull our mouse colony. We were able to keep mice carrying the entire SE deletion.

We now clarify that the 10kb of DNA fulfills the structural definition of a SE and that individual modules might have cell-specific activities (line 105 and lines 372-374). We had already shown that E1 and E2 but not E3 and E4 perform prominent roles in mammary tissue.

As the reviewer suggested, we searched for salivary ATAC-seq and identified one publication (reference 42). We reanalyzed the data but no significant information was obtained (lines 366-367).

Minor concerns:

55. Regarding Supplemental Figure 2: In the text it says that Supplemental Figure 2 shows that only 12 out of the 20 Stat5 binding sites do actually have a GAS motif. That is not shown or indicated anywhere in this figure.

Response

We added motif information for each area covered by transcription factors in Table S2.

56. Supplemental Figure 2a: it is unclear what the distance refers to here. Is this the distance from a promoter? The distance from an enhancer or (as I suspect) the distance from either a promoter or an enhancer? Then showing this in the same figure should be indicated better. Maybe in this case using heatmaps surrounding the enhancers and the promoters is a better way of depicting it? Alternatively, using a gene body distribution of Pol II for example would be a better way of showing these data.

Response

Supplementary Figure 2a shows features of enhancers and promoters for histone markers. Distance 0 is the coordinate of the peaks which were provided as input file and the center of the peak. We added this explanation in the figure legend.

As the reviewer suggested, we prepared heatmaps for the enhancers and promoters instead of coverage plots (below). Like many papers show heatmaps for entire genome, our data show similar heatmaps when we analyzed enhancers and promoters in genome wide (left figure). However, in this study, we focused on the casein locus and heatmaps for the enhancers and promoters in the locus were displayed in right one. We chose to keep the coverage plots in the manuscript, as the heatmaps lack resolution to properly illustrate the difference in histone mark coverage in the Csn locus and are more difficult to interpret by the reader. Since these heatmaps are not efficient to show difference of histone marks in Csn enhancers and promoters, we kept coverage plots in the manuscript.

57. Regarding Figure 2b: the information in this figure is not clear: the whole super-enhancer cluster is deleted, therefore there should not be anything in the samples to be mapped to the region shown in Figure 2b? I suggest using a custom annotation for mapping the reads to the genome, since there should not be a signal in these regions. If the authors want to ensure that the regions is really deleted, then including the Sanger sequencing is a better way to convince a reviewer.

Response

We removed the figure 2b and the Sanger sequencing of mutant mice has been provided in Supplementary Table 13.

58. Figure 2d: shows changes across the whole locus. When the SE is lost, then the individual enhancer at the Csn3 locus also loses H3K27ac. Here a better quantification is necessary (see above). A second conclusion here is that Stat5 binding at the proximal enhancer to Csn1sb2 is lost. Is this Stat5 binding happening at region with a GAS motif? Similar: Supplemental Figure 4: loss of the E2 leads to loss of Stat5 binding. Is that again a real GAS binding event or is it a secondary effect?

Response

To show how much enhancers and promoters' activity is changed in mutant tissues compared to WT, we presented coverage plots in Supplementary Figure 3.

Both sites contain a GAS motif. Table S2 presents information on TF binding motifs detected under each TF peak.

59. Regarding Supplemental Figure 3: B shows the loss of H3K27ac at enhancers. Here it needs to be clarified whether this is at all enhancers or just a certain subset. Also: a control experiment needs to be included looking at enhancers that are not affected, for example from the Wap locus or the Cish locus as presented in this figure.

Response

H3K27ac reduced at all enhancers, but to different extents. As reviewer suggested, we added another milk protein gene, the Wap locus, as a control.

60. Regarding Supplemental Figure 5: It is very interesting that the same SE is also active early in pregnancy and that multiple genes are downregulated at the analyzed time points. The authors claim that loss of the SE affects the initiation of casein enhancers. From the figure shown this is really hard to judge: more rigorous quantifications and normalizations are needed, comparison to a non-affected locus should be included that is specifically expressed at this point of development. Which enhancers are affected in the opinion of the authors?

Response

RNA-seq data show that all Csn genes are affected by the SE at day 6 of pregnant. As a control, we added Wap locus. At this point the mechanism is not clear, and it could occur at different levels.

61. Line 131/132 and discussion: the comment about the distance dependency about the SE would be good to be discussed in the light of the two recent publications from the Giorgetti lab (Zuin et al, Nature 2022) and de Laat group (Rinzema et al., NSMB, 2022). Line 149/150: Interactions between the Csn3 enhancers and the SE had been confirmed by 3C, further supporting their crosstalk. How does this crosstalk look like? Loss of the E2 enhancer, so the promoter proximal enhancer, abrogates most transcription. Does that now mean that the SE is not needed for the activation of the Csn3 gene? Or that the E2 element is needed for the activation of both E1 and the SE on the gene?

Response

We thank the reviewer for pointing out the studies by Zuin and Rinzema, which now included in the discussion. Regarding the Csn3 gene, the presence of SE is essential for the activation of Csn3 expression (Figure 2).

62. Line 189: “additive activity in the alpha globin locus”. Upon reexamination of the alpha globin locus in a recent Biorxiv paper, Higgs and colleagues have shown that it is not just simple additivity that works at the locus, but that some elements are needed for the locus to fully activate even though they do not have their own activity (Blayney et al, BioRxiv, 2022).

Response

We thank the reviewer for pointing out this publication. We discussed these findings in the context of our study. We also discussed our work in the context of another paper from the Higgs group (Kassouf et al) that shows orientation dependency of the α -globin SE. (lines 348-354).

REVIEWER COMMENTS

Reviewer #1 (Remarks to the Author):

In this revised manuscript, now re-titled “Cell-specific and shared regulatory elements control a multigene locus active in mammary and salivary glands”, Lee et al. have carried out combinatorial inactivation of regulatory elements within a locus that encodes 8 genes relevant to either exclusively in either mammary and salivary glands or in both tissues. They have specifically identified a super-enhancer, gene-specific local enhancers and a cytokine-responsive promoter within the locus and highlight, through their experiments, a certain level of functional redundancy and complexity between them. Since the last submission, the authors have improved the paper significantly by introducing more explanations of the rationale in the writing and more data. Besides data, they had already generated several deletion mutants for their mouse genetics experiments, which would be great resource for the community. Besides being technically beneficial, this study also provides insights into the complexity of gene regulation as observed of a large multi-gene locus expressed in two different epithelial tissues. They have responded well to both the major and minor concerns that we had raised previously.

Reviewer #2 (Remarks to the Author):

In this revised manuscript the authors describe an extensive study of the regulatory landscape of the casein gene locus which contains genes expressed in epithelial cells, 5 casein genes expressed predominantly and at high levels in the lactating mammary gland epithelium and 3 genes expressed predominantly in the salivary gland epithelium among other tissues, one of which is also detected in the lactating mammary gland. This is a real tour the force with the sheer number of deletion-mutant mice analyzed for different aspects of transcription factor binding, chromatin marks and expression. It gives fascinating insights in the complexity of the regulation of a large multi gene locus predominantly expressed in two different epithelial tissues. It makes clear that both larger Super-Enhancers and smaller locally acting enhancers and promoters are important to achieve full functionality, and that some level of redundance might exist to safeguards a function as important for species survival as lactation.

The authors have mostly addressed concerns brought up in the initial review, they have now included Csn2-enhancer data previously reported in a separate manuscript and added more data such as DNA methylation data and other information, and discuss their findings more in the context of previous work.

The inclusion of the Csn2 data helps provide a more complete picture of the complex regulation of this gene locus. However, I still fail to grasp how the Csn2 promoter and enhancers are different from the Csn1s1-E2 and promoter (both binding STAT5) the Csn 3 promoter/E and CSN1s2b promoter and E (Previously published). Stat5 is bound at all Csn gene promoters and most enhancers in lactation so this should be discussed in a more general context of all casein genes. No point in singling out Csn2.

With that, line22/23 of the abstract (>10000 fold induction) is applicable to all casein genes and should be stated as such. Same for line 62/63.

The table (S14) indicating at which timepoints mammary gland tissue was collected and the data sets that were re-used and generated for this manuscript is very helpful, and makes clear the tour the force in generating and analyzing mutant mouse lines associated with this manuscript.

To make this data more generally accessible for readers and other researchers, I suggest creating a track hub for UCSC genome browser where readers can access all tracks associated with this manuscript. This would also provide a way to view the data in more detail and clearer than the current figures, as the differences in coverage are hard to see in some cases in the genome browser view images.

Previously suggested to include coverage profiles for the individual regulatory regions discussed in the manuscript for WT vs Mut and the different time points (P6 P18, L1, L10), similar to the coverage plots in Lee et al 2018, coverage trace-difference changes in binding/enrichment will help to clearly visualize the changes at the different genes and enhancers discussed. The current aggregate coverage plots in S2 and S3 do not visualize this.

Giving coordinates on browser for each of the figures would also aid in viewing of the data.

In table S14 on the harvested tissue list WT-SG is listed twice probably should be Δ SE-SG.

Could you please explain why salivary gland from male mice were used. This matters as some of the activity could be hormone dependent and different in females and males. Furthermore, there is no mention in the methods how RNA was isolated for RNA-seq or qRT-PCR for salivary gland tissue.

The description of the mammary tissue collected in line 463/464 does not make sense: "Mammary tissues from pregnant females with 2.5-3 month-old age at specific stages (virgin, pregnancy day 6, lactation day 1 and 10)" pregnant mice are not virgin nor lactate. Were the virgin mice staged for estrus-cycle?

Furthermore, there are still numerous instances in the figure legends where it is not clear which timepoint is depicted. Please refrains from saying lactation or pregnancy, as for certain mutant both L1 and L10 and pregnancy day 6 and 18 where collected and one cannot expect the reader to memorize the table.

Fig 1: B & C .I assume it is L1 for C as no L10 ChIP-Seq data use was listed but B could be L1 or L10

Also Figure 1

1C:it appears that the data depicted and the labeling of the genes for Cns2, Csn1s2a and Csn1s2b are incorrect (my guess: data for Csn1s2a labeled as Csn2, Csn1s2b data labeled as Csn1s2a and Csn2 data labeled as Csn1s2b) while fixing this might as well include the pregnancy day 18 RNA-seq data for a complete depiction, especially as it might be illuminating for readers to understand that major gene induction takes place at the end of pregnancy for most caseins but Csn1s2b.

Fig 2: C, " during pregnancy". Might be pregnancy day 18 (although no datasets for day 18 are listed in Tab S14)

Fig 3: A) No need to break in Y-axis, mention that WT and delta SE is analyzed and shown. B) does not list the Mammary gland shown nor its stage...Twice mention: "The red shade indicates the super-enhancer"

Fig 4: C) lactating mammary gland? L1?

Fig 5: A) Stage/ time point C) lactating mammary gland? L1?

Fig6: B) lactating? L1? C) which timepoint? >> GR distribution seems changed

Fig 7: C) lactating/ L1?

Fig 8: C) lactating/ L1?

Figure 9: A) stages?

heatmap column labels: delta E1/2/2 .. > Delta E1/2/3. Is Csn1s2b-deltaE1 correct label? Is this from previous paper? If so what does this refer to in the previous paper the label E1 is not used.

Suspect that for the second column data and labels are not correctly associated (as in figure 1c).

Suggest picking a color scheme for the heat map that makes it easier to distinguish differences especially above 50%.

why not include Csn1s1 mutants?

Need more details in the legend of this figure why those mutants included, if they are from this manuscript or earlier (with reference)

Might be worth to also include data for different time points to give a better appreciation for the temporal aspects, and possibly including the intra genic enhancer for Csn1s2b.

Supplemental figures

Fig S3: timepoint?

Fig S4: a) timepoint?

Fig S5: label the tow E2 as CSn1s1-E2 and CSN3-E2

Fig S6: timepoint?

Fig S9: A &C) timepoint? B) lactation: L1 or L10? Sorted cell are mentioned how were they isolated and sorted? This is not reported in the methods.

Please indicate for table S9, 10 and 11 that this pertains to Csn2 promoter enhancer analysis.

Csn3

Effect on K27Ac and Pol2 seems minimal except maybe for SMr3a, little correlation to expression changes.

Line 130-133: females with SE deletion fail to lactate and it is stated that all analysis have been performed at day 18 of lactation because "This was critical as lactation failure results in rapid loss of cell differentiation and tissue

remodeling after parturition". However, from Table S14 and my reading of the manuscript it appears that all Chip-Seq was performed at day 1 of lactation (2-3days later than RNA-seq) so this makes the connection and comparison of expression (at P18) and chromatin organization and TF binding (at L1) a bit fraud by the authors own admission.

Line 133-137: as the Csn2 gene analysis are now incorporated in this manuscript the results for Csn2 and Csn1s2a should be mentioned here too.

Line 179-180: the deletion of SE-E1/2 mimics deletion of 10kb SE fragment (assuming this relates to gene expression) does this also result in lactation failure? I notice that it is indicated that the qRT-PCR was performed in L1 tissue.

Line232: "...had been confirmed..." >have been confirmed...

Line 247: "...reduction of Csn1s1 these mice.." >in these mice...

Line 271-274: "While TF binding at enhancers was within a H3K27ac 'gap', binding at promoters was outside a classical gap. This points to the possibility that TF binding at enhancers and promoters and their interactions with H3K27ac marks is distinctly different and might serve different purposes" it might not be a classical gap because the K27ac coverage over the gene that is being transcribed is lower in the transcribed promoter. A less transcription factor centered interpretation could be that there is a different chromatin organization due to the gene transcription or 3D organization of these genes. (How does this look at other csn genes at L1?) I suggest deleting this.

Line 302-305, Figure S8 Table S10:

No statistics are provided for figure S8 for virgin Csn2 Prom and enhancer deletions or for the lactation day 10 data. Supplementary table 11 suggests that Csn 2 and Cn1s2b are significantly up regulated as are several genes involved in lipid/FA synthesis, while RankL is the most down regulated gene in the P/E123 deleted mice. Please address this ???

Although the authors have referenced some of the previous work related to Csn2-E1 (BCE-1, beta casein enhancer) they still fail to reference that this region was identified in cow and human genome more than 30 years ago this should be referenced > PMID: 1498370 PMID: 9528790 PMID: 9795185

Line 338/339: "... and three gained independence of the SE...at least in lactation" this is an over statement and simplification of the findings. Partial independence would be more appropriate as the expression is still affected at a significant level just not as drastically as others. Besides the expression analysis for the full deletion where presumably done at P18 (not lactation) and the composite deletions at L1 (despite lactation failure???, see previous comments). Think the interpretation needs to be more nuanced here.

Line 367/368: "Key TFs controlling mammary function, such as STAT58,9 and ELF57 are dispensable in the salivary gland43." How about GR? not GR would be my guess.

Line 369: "...but its role in salivary..." > salivary

Cytokine responsive gene promoters:

I still fail to see why Csn2 needs to be singled out in this context all casein genes are regulated to some level by STAT5 and STAT5 sites are present and STAT5 binds at promoter and enhancers. Furthermore, the casein gene cluster is rife with enhancers locally acting and distally (that was already established to some level before this detailed in vivo analysis of the gene cluster regulations). So the importance of discussing the casein genes in the context of epromoters escapes me. IMO, this should be left out and instead discuss all promoters (including previous published work on Csn1s2b) in the locus and their dependence on enhancers and STAT5 and other TFs. Some attempt to put all this in the context of the temporal changes with pregnancy and lactation seems more appropriate.

Line 294/295: "Although both Epromoters and the five Casein gene promoters, including Csn2 investigated in this study..." seems the Authors did copy this from integrated manuscript did not do due diligence to properly vet text to make it truly integrate into the manuscript.

There are still some methods descriptions that only list one tissue and/or time point, please mind the details of methods.

Line 555: "Identification of regulatory elements:..." it is probably more accurate to title this regulatory elements bound by STAT5 in lactation.

Line 595/596: "Data set used in the study was listed in Supplementary Table 14." > Datasets used....are listed...

Reviewer #3 (Remarks to the Author):

Overall, the study has much improved in the revision process and the suggestion to combine both study into one bigger study is very successful. I would have liked to see more coherence and integration of the results from the previous two studies into the one manuscript now, for example, results from one gene, here *Csn1s1*, could then also be discussed in the context of another gene, here *Csn2*, since in both cases, the Stat5 binding site at the promoter plays a prominent role however, only for the latter that is then shown.

There are a few proposals that the authors should address in writing, since they make very strong statements without considering alternatives (see below). All in all though, this is a tour de force and a very strong candidate for publication in Nature Communications.

Line 250/250: The authors discuss the effect of the loss of the individual enhancers at the *Csn1s1* gene and suggest that they have limited biological activity and that the principal regulator of *Csn1s1* expression in pregnancy lies within the Stat5 of the promoter. Alternatively, the two enhancers could be partially redundant and work in a superadditive fashion, so that only when both enhancers are removed, the effect on *Csn1s1* expression would be revealed.

Lines 272/273/274: The authors suggest that there might be differences in the promoter and enhancer binding of the transcription factors, since they do not observe a classical gap or nucleosome free region at the promoter. At the resolution shown, this is not possible to judge, alternatively, the promoter region could be more flexible and show a higher heterogeneity in the TF binding. In a population wide assay such as ChIP-seq this is hard to judge without further data.

We thank the reviewers for their valuable feedback, which has greatly contributed to enhancing the clarity and impact of our manuscript. Their expertise and attention to detail have been invaluable in shaping the final version of the manuscript. By carefully considering their recommendations, we believe our manuscript has been significantly improved.

Our responses to the reviewers' comments are marked in red.

Reviewer #1 (Remarks to the Author):

In this revised manuscript, now re-titled “Cell-specific and shared regulatory elements control a multigene locus active in mammary and salivary glands”, Lee et al. have carried out combinatorial inactivation of regulatory elements within a locus that encodes 8 genes relevant to either exclusively in either mammary and salivary glands or in both tissues. They have specifically identified a super-enhancer, gene-specific local enhancers and a cytokine-responsive promoter within the locus and highlight, through their experiments, a certain level of functional redundancy and complexity between them. Since the last submission, the authors have improved the paper significantly by introducing more explanations of the rationale in the writing and more data. Besides data, they had already generated several deletion mutants for their mouse genetics experiments, which would be great resource for the community. Besides being technically beneficial, this study also provides insights into the complexity of gene regulation as observed of a large multi-gene locus expressed in two different epithelial tissues. They have responded well to both the major and minor concerns that we had raised previously.

Response

We thank the reviewer for the positive evaluation.

Reviewer #2 (Remarks to the Author):

In this revised manuscript the authors describe an extensive study of the regulatory landscape of the casein gene locus which contains genes expressed in epithelial cells, 5 casein genes expressed predominantly and at high levels in the lactating mammary gland epithelium and 3 genes expressed predominantly in the salivary gland epithelium among other tissues, one of which is also detected in the lactating mammary gland. This is a real tour the force with the sheer number of deletion-mutant mice analyzed for different aspects of transcription factor binding, chromatin marks and expression. It gives fascinating insights in the complexity of the regulation of a large multi gene locus

predominantly expressed in two different epithelial tissues. It makes clear that both larger Super-Enhancers and smaller locally acting enhancers and promoters are important to achieve full functionality, and that some level of redundancy might exist to safeguard a function as important for species survival as lactation.

Response

We thank the reviewer for the positive evaluation.

The authors have mostly addressed concerns brought up in the initial review, they have now included Csn2-enhancer data previously reported in a separate manuscript and added more data such as DNA methylation data and other information and discuss their findings more in the context of previous work.

The inclusion of the Csn2 data helps provide a more complete picture of the complex regulation of this gene locus. However, I still fail to grasp how the Csn2 promoter and enhancers are different from the Csn1s1-E2 and promoter (both binding STAT5) the Csn 3 promoter/E and CSN1s2b promoter and E (Previously published). Stat5 is bound at all Csn gene promoters and most enhancers in lactation so this should be discussed in a more general context of all casein genes. No point in singling out Csn2. With that, line22/23 of the abstract (>10000-fold induction) is applicable to all casein genes and should be stated as such. Same for line 62/63.

Response

We thank the reviewer for raising the very important issue of potentially regulatory differences between the five Casein genes. Since this question had not been fully addressed in the previous version of the manuscript, we now added a paragraph, called 'Regulation of the five Casein genes' in the discussion section (lines 398 – 437). As we have shown in this study, expression of the Csn2 gene is dependent on the combined presence of three distal enhancers and the non-canonical STAT5 binding site (GAS motif) within promoter. The Csn1s1 gene contains two distal enhancers, which we deleted individually, and the biological consequences were restricted. It is therefore well possible that, like the Csn2 gene, both enhancers and the STAT5 site in the promoter are required for the activation of the Csn1s1 gene during pregnancy and lactation. We stated these possibilities within the Result part (lines 257 – 261) and the Discussion (lines 416 – 419).

Regulation of the Csn1s2b gene is quite distinct from the other Casein genes as its main activation coincides with parturition. This is likely controlled, at least in part, by a unique intronic regulatory element we reported in a previous study. The regulatory elements controlling Csn3 expression have distinct features not seen in the other casein

genes. For example, the Csn3 promoter does not harbor a STAT5 binding site and is therefore not under STAT5 control. Moreover, only one of the upstream enhancers is functional by itself. We addressed these points on the new paragraph in the Discussion (lines 420 – 430).

We agree with the reviewer that Csn2 should not be singled out and we therefore modified the Abstract as follows: “Casein genes (Csn1s1, Csn2, Csn1s2a, Csn1s2b, Csn3) are expressed in mammary glands, induced 10,000-fold during pregnancy” (lines 16 – 17).

The table (S14) indicating at which timepoints mammary gland tissue was collected and the data sets that were re-used and generated for this manuscript is very helpful, and makes clear the tour the force in generating and analyzing mutant mouse lines associated with this manuscript.

Response

We thank the reviewer for appreciating this study and the efforts going into it.

To make this data more generally accessible for readers and other researchers, I suggest creating a track hub for UCSC genome browser where readers can access all tracks associated with this manuscript. This would also provide a way to view the data in more detail and clearer than the current figures, as the differences in coverage are hard to see in some cases in the genome browser view images.

Previously suggested to include coverage profiles for the individual regulatory regions discussed in the manuscript for WT vs Mut and the different time points (P6 P18, L1, L10), similar to the coverage plots in Lee et al 2018, coverage trace-difference changes in binding/enrichment will help to clearly visualize the changes at the different genes and enhancers discussed. The current aggregate coverage plots in S2 and S3 do not visualize this.

Giving coordinates on browser for each of the figures would also aid in viewing of the data.

Response

For this study we generated and analyzed a total of 167 new NGS data sets (124 ChIP-seq, 38 RNA-seq and 5 methyl-seq) that have been deposited in GEO. The reviewer suggested to create a track hub for UCSC genome browser. We believe that creating a

track hub containing more than 170 ChIP-seq data sets (128 from the current study and more than 40 from previous studies) would be unwieldy. We believe that downloading the processed data sets from GEO would be a better and more feasible approach. We have uploaded both the raw data files and the processed files that can be downloaded and imported into the IGV browser without difficulties. We have now included detailed instructions in the M&M section (lines 671 – 675).

Processed bedGraph files deposited in GEO (Gene Expression Omnibus) can be visualized using the Integrative Genomics Viewer (IGV) browser with a reference genome. We stated the following instructions in the M&M section.

To visualize bedGraph files in IGV, these general steps should be followed:

- Download the processed ChIP-seq data with bedGraph format from GEO (<https://www.ncbi.nlm.nih.gov/gds/?term=>).*
- Download IGV (<https://software.broadinstitute.org/software/igv/download>) and launch it on your computer.*
- Select mm10 reference genome and go to the chromosome of interest.*
- Load the processed ChIP-seq file like bam, bedGraph, Wig or bigWig format.*
- Search regions you are interested in.*

In table S14 on the harvested tissue list WT-SG is listed twice probably should be Δ SE-SG.

We newly generated coverage plot and bar graph to each regulatory elements and added them in Supplementary Figure 3. In bar graph, we compared ChIP-seq activity using counting read numbers and normalizing them with one of control locus and WT data set.

In table S14 on the harvested tissue list WT-SG is listed twice probably should be Δ SE-SG.

Response

We thank the reviewer for finding the typo. We corrected it.

Could you please explain why salivary gland from male mice were used. This matters as some of the activity could be hormone dependent and different in females and males. Furthermore, there is no mention in the methods how RNA was isolated for

RNA-seq or qRT-PCR for salivary gland tissue.

Response

We used male salivary glands as this avoids the contamination with mammary tissue. As the reviewer knows, the cervical (#1) mammary gland in female mice encroaches into the salivary region and it is virtually impossible to dissect salivary tissue without contaminating mammary tissue. Since casein expression in mammary tissue is several orders of magnitude higher than that in salivary tissue, even a contamination of less than 1% would have severe consequences and render the experiment uninterpretable.

We have included instructions in the M&M section (lines 539 – 540, 582 – 583).

The description of the mammary tissue collected in line 463/464 does not make sense: “Mammary tissues from pregnant females with 2.5-3 month-old age at specific stages (virgin, pregnancy day 6, lactation day 1 and 10)” pregnant mice are not virgin nor lactate. Were the virgin mice staged for estrus-cycle?

Response

We apologize for the lack of clarity and the corrected sentence as follows “Two-months old mice were used in the experiments. Mammary tissue was collected from non-parous (virgin) mice, from mice at day 6 of pregnancy and at days 1 (L1) and 10 (L10) of lactation and stored at -80°C” (lines 531 – 533). No, the virgin mice were not staged for the estrous cycle.

Furthermore, there are still numerous instances in the figure legends where it is not clear which timepoint is depicted. Please refrains from saying lactation or pregnancy, as for certain mutant both L1 and L10 and pregnancy day 6 and 18 where collected and one cannot expect the reader to memorize the table.

Fig 1: B & C .I assume it is L1 for C as no L10 ChIP-Seq data use was listed but B could be L1 or L10

Response

We added the specific time points of RNA-seq and ChIP-seq data in the figure legend.

Also Figure 1

1C: it appears that the data depicted and the labeling of the genes for Cns2, Csn1s2a

and Csn1s2b are incorrect (my guess: data for Csn1s2a labeled as Csn2, Csn1s2b data labeled as Csn1s2a and Csn2 data labeled as Csn1s2b) while fixing this might as well include the pregnancy day 18 RNA-seq data for a complete depiction, especially as it might be illuminating for readers to understand that major gene induction takes place at the end of pregnancy for most caseins but Csn1s2b.

Response

We thank the reviewer for pointing out the oversight and the results and gene names have been corrected. In our experience, expression of individual milk protein genes is similar between pregnancy day 18.5 (p18) and lactation day 0.5 (within 12 hours post-partum (pp <12h), called as L1) in C57BL/6N mice. Therefore, we used lactation day 1 RNA-seq data from WT mice in the figure. As the reviewer knows, the morning the vaginal plug is detected is called '0.5 days of pregnancy'.

Fig 2: C, " during pregnancy". Might be pregnancy day 18 (although no datasets for day 18 are listed in Tab S14)

Response

We added the specific time point of ChIP-seq data in the figure legend and the sentence now reads "Mammary tissues of WT and Δ SE mice were collected within 12 hours post-partum (pp <12h)".

Fig 3: A) No need to break in Y-axis, mention that WT and delta SE is analyzed and shown. B) does not list the Mammary gland shown nor its stage... Twice mention: "The red shade indicates the super-enhancer"

Response

We edited the Y-axis of figure A and added specific time point of mammary gland ChIP-seq data and explanation for the red shade in the figure legend.

Fig 4: C) lactating mammary gland? L1?

Response

We added specific time point of mammary gland ChIP-seq data in the figure legend. It now states: "Genomic features of the Odam-Csn3 locus were characterized by ChIP-seq in lactating mammary (day one of lactation, L1)".

Fig 5: A) Stage/ time point C) lactating mammary gland? L1?

Response

We added specific time point of ChIP-seq data in the figure legend. It now states: “the presence of H3K27ac and H3K4me1 marks in mammary glands of wild type mice at day one of lactation (L1) indicate a distal candidate enhancer (Csn3-E1) at -7 kbp and a proximal one (Csn3-E2) at -0.6 kbp”.

We now stated in the M&M and the figure legend that mammary tissue was collected within 12 hours post-partum (pp <12h). (lines 533 – 539, 961)

Fig6: B) lactating? L1? C) which timepoint? >> GR distribution seems changed

Response

The data were generated from mammary tissue collected at day one of lactation (L1) and we added this information in the Figure legend.

It is not clear what the reviewer means by “GR distribution seems changed”.

Fig 7: C) lactating/ L1?

Response

The ChIP-seq data were generated from mammary tissue collected at day one of lactation (L1) and we added this information in the figure legend.

Fig 8: C) lactating/ L1?

Response

The data were generated from mammary tissue collected at day one of lactation (L1) and we added this information in the figure legend (line 1005).

Figure 9: A) stages?

heatmap column labels: delta E1/2/2 .. > Delta E1/2/3. Is Csn1s2b-deltaE1 correct label? Is this from previous paper? If so what does this refer to in the previous paper the label E1 is not used.

Response

We thank the reviewer for pointing out the typo and gene name. We corrected their names.

Suspect that for the second column data and labels are not correctly associated (as in figure 1c).

Response

The figure A is a diagram to show relative gene expression level of genes in the casein locus between WT and mutant mice. The data are correct.

Suggest picking a color scheme for the heat map that makes it easier to distinguish differences especially above 50%.

Response

As the reviewer suggested, we used 10 different colors for each 10% to distinguish differences of gene expression.

Why not include Csn1s1 mutants?

Response

The data of heatmap are from RNA-seq data. However, we didn't conduct RNA-seq with mammary tissue from Csn1s1 mutant mice because we did not see any significant difference of Csn1s1 gene expression in the mutant mice as compared to WT mice using qRT-PCR.

Need more details in the legend of this figure why those mutants included, if they are from this manuscript or earlier (with reference)

Might be worth to also include data for different time points to give a better appreciation for the temporal aspects, and possibly including the intra genic enhancer for Csn1s2b.

Response

All Casein genes are highly expressed at day 18 of pregnancy (p18) and day 1 of lactation (L1) as compared to day 6 of pregnancy (p6). Since mutant mice lacking the

super-enhancer (ΔSE) and mice lacking the Csn3 enhancer (Csn3- $\Delta E2$) failed to lactate and were unable nurse their pups (likely the result of an absence of CSN3), we could not generate data from day 10 of lactation (L1). Therefore, we focused on day 18 of pregnancy and day 1 of lactation (or post-partum >12 hours in case of non-lactating mice).

We added additional information, and the text figure legend reads: “The data from the Csn2, Csn3 and ΔSE mice are from this study and the Csn1s2b mutant has been reported earlier²⁵” (lines 1012 – 1013).

Supplemental figures

Fig S3: timepoint?

Fig S4: a) timepoint?

Fig S5: label the tow E2 as CSn1s1-E2 and CSN3-E2

Fig S6: timepoint?

Response

We added specific time point in the figure legend.

Fig S9: A & C) timepoint? B) lactation: L1 or L10? Sorted cell are mentioned how were they isolated and sorted? This is not reported in the methods.

Response

We added specific time points in the figure legend. We mentioned sorted cells in Fig S6 and data from sorted data are from reference 23. The figure legend reads: “Csn2 mRNA levels in lactating mammary tissues from WT and mutant mice at day one of lactation (L1) were measured by qRT-PCR and normalized to Gapdh levels.”

Please indicate for table S9, 10 and 11 that this pertains to Csn2 promoter enhancer analysis.

Response

We indicated clearly Csn2 mutant lines in figure legend. The figure legend reads: “Supplementary Table 9. List of all genes with normalized read counts in each replicate at L1 mammary tissue between WT and Csn2- $\Delta E1/2/3$, Csn2- ΔP or Csn2-P-E1/2/3 mutants, \log_2 (fold change), their p-value and adjusted p-value.

Supplementary Table 10. List of all genes with normalized read counts in each replicate at L10 mammary tissue between WT and Csn2- Δ E1/2/3, Csn2- Δ P or Csn2-P-E1/2/3 mutants, \log_2 (fold change), their p-value and adjusted p-value.

Supplementary Table 11. List of all genes with normalized read counts in each replicate at virgin mammary tissue between WT and Csn2- Δ P or Csn2-P-E1/2/3 mutants, \log_2 (fold change), their p-value and adjusted p-value.”

Csn3

Effect on K27Ac and Pol2 seems minimal except maybe for SMr3a, little correlation to expression changes.

Line 130-133: females with SE deletion fail to lactate and it is stated that all analysis have been performed at day 18 of lactation because “This was critical as lactation failure results in rapid loss of cell differentiation and tissue remodeling after parturition”. However, from Table S14 and my reading of the manuscript it appears that all Chip-Seq was performed at day 1 of lactation (2-3days later than RNA-seq) so this makes the connection and comparison of expression (at P18) and chromatin organization and TF binding (at L1) a bit fraud by the authors own admission.

Response

We need to clarify that the gestation period of our mouse strain (C57BL/6) is 18.5 days (http://ko.cwru.edu/info/breeding_strategies_manual.pdf and PMID: 20811634). Since the mammary tissue for the RNA-seq had been collected at p18 and the tissue for ChIP-seq within 12 hours of parturition (post-partum <12 hours) later it is fair to integrate the data.

Importantly, the ChIP-seq patterns at genes not affected by the enhancer deletion, were equivalent in post-partum tissue from wild type and mutant mice.

We clarify the exact time windows when tissue was harvested, we expanded on the M&M section (lines 531 – 539). “Two-months old mice were used in the experiments. Mammary tissue was collected from non-parous (virgin) mice, from mice at day 6 of pregnancy and at days 1 (L1) and 10 (L10) of lactation and stored at -80°C. While most mutations did not affect the ability of dams to lactate and nurse their pups, deletion of the super-enhancer (Δ SE and Δ SE1/2) and deletion of the Csn3 E2 enhancer (Csn3- Δ E2) resulted in lactation failure, which is most likely due to the absence of CSN3. The C57Bl/6 mouse strain used in this study has a gestation period of 18.5 days²¹ and mammary tissues from mutant mice unable to lactate (Δ SE, Δ SE1/2 and Csn3- Δ E2)

were collected at day 18 of pregnancy (p18) or within 12 hours after delivery (post-partum <12 hours, pp <12h)”.

As the reviewer knows, it takes 24 hours for STAT5 phosphorylation to disappear in mice that cannot lactate, and we are well within the time period of lactation reversal as we described in an earlier study (Li et al. 1997. PMID: 9096410).

We also rephrased the result part as follows and added references. “Female mice lacking the SE were unable to support their litters due to lactation failure and analyses were conducted in mammary tissue at day 18 of pregnancy, just prior to delivery or within 12 hours after delivery (post-partum <12 hours, pp <12h). This was critical as lactation failure results in the loss of cell differentiation and loss of STAT5 phosphorylation and tissue remodeling is observed between 12 and 24 hours after parturition in a mouse model^{P0}. Of note, the C57BL/6 strain used in this study has a gestational period of 18.5 days²¹” (lines 130 – 136).

Line 133-137: as the Csn2 gene analysis are now incorporated in this manuscript the results for Csn2 and Csn1s2a should be mentioned here too.

Response

We thank the reviewer for pointing this out and we now added the following sentence “Csn1s1, Csn2 and Csn1s2a expression was preserved at approximately 50%, 70% and 60% of wild-type (WT) levels, respectively (lines 141 – 142)”.

Line 179-180: the deletion of SE-E1/2 mimics deletion of 10kb SE fragment (assuming this relates to gene expression) does this also result in lactation failure? I notice that it is indicated that the qRT-PCR was performed in L1 tissue.

Response

Yes, mice carrying the SE-E1/2 deletion also fail to lactate, and we had stated that this mutation mimicked the total super-enhancer deletion. We supported this with an additional sentence “Of note, Δ E1/2 mice exhibited lactation failure” (lines 185 – 186).

Line232: “...had been confirmed...” >have been confirmed...

Line 247: “...reduction of Csn1s1 these mice..” >in these mice...

Response

We corrected the grammar.

Line 271-274: “While TF binding at enhancers was within a H3K27ac ‘gap’, binding at promoters was outside a classical gap. This points to the possibility that TF binding at enhancers and promoters and their interactions with H3K27ac marks is distinctly different and might serve different purposes” it might not be a classical gap because the K27ac coverage over the gene that is being transcribed is lower in the transcribed promoter. A less transcription factor centered interpretation could be that there is a different chromatin organization due to the gene transcription or 3D organization of these genes. (How does this look at other csn genes at L1?)
I suggest deleting this.

Response

We agree with the reviewer that there is further room for different interpretations, and we included the following sentence “It is further possible that promoter regions are more flexible and present higher heterogeneity in TF binding and different chromatin organization due to the gene transcription or 3D organization of these genes can be envisioned” (lines 284 - 287).

Line 302-305, Figure S8 Table S10:

No statistics are provided for figure S8 for virgin Csn2 Prom and enhancer deletions or for the lactation day 10 data. Supplementary table 11 suggests that Csn 2 and Cn1s2b are significantly up regulated as are several genes involved in lipid/FA synthesis, while RankL is the most down regulated gene in the P/E123 deleted mice. Please address this ???

Response

We would like to emphasize that the purpose of our study was the investigation of the impact of enhancer and promoter elements in pregnancy and lactation-induced casein gene expression and the virgin data were a byproduct. We hope that the extensive data sets from virgin mice will be useful to other researchers in the field.

It is not clear why Csn2 levels in mutant mice are higher than in WT mice (564 vs 313 reads). The estrous cycle certainly affects gene expression in non-parous mice as we have shown earlier (Robinson et al. 1995, PMID: 7635053). As the reviewer is aware, RANKL is regulated by TNF-alpha, which is also reduced in the Csn2 promoter-enhancer mutants.

Although the authors have referenced some of the previous work related to Csn2-E1 (BCE-1, beta casein enhancer) they still fail to reference that this region was identified in cow and human genome more than 30 years ago this should be referenced > PMID: 1498370 PMID: 9528790 PMID: 9795185

Response

We have now referenced these papers that have identified sequences that respond to prolactin and extracellular matrix in studies using reporter gene assays in cell transfection experiments. We have also pointed out differences observed between our mouse genetic studies and the transfection studies in cell lines on the Discussion (lines 477 – 485).

Line 338/339: "... and three gained independence of the SE...at least in lactation" this is an over statement and simplification of the findings. Partial independence would be more appropriate as the expression is still affected at a significant level just not as drastically as others. Besides the expression analysis for the full deletion where presumably done at p18 (not lactation) and the composite deletions at L1 (despite lactation failure???, see previous comments). Think the interpretation needs to be more nuanced here.

Response

We agree with the reviewer and have toned down the interpretation. We now state: "However, as this locus expanded, the five newly formed Casein genes acquired their own regulatory elements and three gained partial independence of the SE, at least during lactation" (lines 353 – 355).

Line 367/368:" Key TFs controlling mammary function, such as STAT58,9 and ELF57 are dispensable in the salivary gland43." How about GR? not GR would be my guess.

Response

The literature on GR and mammary development and lactation and a bit checkered but it appears that the GR is not needed for 'normal' mammary development (PMID: 11581013).

Line 369: "...but its role in salary..." > salivary

Response

We corrected the typo.

Cytokine responsive gene promoters:

I still fail to see why Csn2 needs to be singled out in this context all casein genes are regulated to some level by STAT5 and STAT5 sites are present and STAT5 binds at promoter and enhancers. Furthermore, the casein gene cluster is rife with enhancers locally acting and distally (that was already established to some level before this detailed in vivo analysis of the gene cluster regulations). So the importance of discussing the casein genes in the context of epromoters escapes me. IMO, this should be left out and instead discuss all promoters (including previous published work on Csn1s2b) in the locus and their dependance on enhancers and STAT5 and other TFs. Some attempt to put all this in the context of the temporal changes with pregnancy and lactation seems more appropriate.

Response

As stated earlier in this response, we did not intend to single out Csn2, it is just the one that has been investigated to the largest extent. Having said this, Csn2 regulation is distinct from Csn1s2b and Csn3. While a single distal enhancer controls Csn1s2b expression, the Csn2 distal enhancers play a modulating role. Similarly, the Csn3 enhancer plays a more dominant role. What we know is that Csn2 expression depends on a synergy between three distal enhancers and a non-canonical STAT5 binding site within the promoter.

We believe that the epromoter discussion is highly relevant as it points to interferon-responsive genes that use STAT1/2 to activate promoters and neighboring genes, which is distinct from the Csn2 situation.

We now added a paragraph in the discussion comparing the regulation of the different caseins and the underlying elements (lines 398 – 437).

Line 294/295: "Although both Epromoters and the five Casein gene promoters, including Csn2 investigated in this study..." seems the Authors did copy this from integrated manuscript did not do due diligence to properly vet text to make it truly integrate into the manuscript.

Response

The reviewer's comment is not clear.

There are still some methods descriptions that only list one tissue and/or time point, please mind the details of methods.

Line 555: "Identification of regulatory elements:..." it is probably more accurate to title this regulatory elements bound by STAT5 in lactation.

Response

We thank the reviewer for suggesting the title: "Identification of regulatory elements bound by STAT5 in lactation". We edited it accordingly.

Line 595/596: "Data set used in the study was listed in Supplementary Table 14." > Datasets used....are listed...

Response

We edited the sentence with reviewer's suggestion (line 671).

Reviewer #3 (Remarks to the Author):

Overall, the study has much improved in the revision process and the suggestion to combine both study into one bigger study is very successful. I would have liked to see more coherence and integration of the results from the previous two studies into the one manuscript now, for example, results from one gene , here Csn1s1, could then also be discussed in the context of another gene, here Csn2, since in both cases, the Stat5 binding site at the promoter plays a prominent role however, only for the latter that is then shown.

Response

We agree with the reviewer that it is important to elaborate on the similarities and differences in the regulation of the five casein genes. In our revised manuscript, we have provided a detailed discussion of these aspects and added a paragraph entitled "Regulation of the five casein genes" (lines 398 – 437). Specifically, we focused on the differential role of the super-enhancer in controlling the five casein genes, the promoter

sequence, the local gene-specific enhancers and the Csn1s2b intronic regulatory element that controls temporal expression.

There are a few proposals that the authors should address in writing, since they make very strong statements without considering alternatives (see below). All in all, though, this is a tour de force and a very strong candidate for publication in Nature Communications.

Response

We very much appreciate the positive comments. Yes, this study was a tour de force, spanning six years of intensive and focused research that involved the generation and analysis of 21 new lines of mutant mice.

Line 250/250: The authors discuss the effect of the loss of the individual enhancers at the Csn1s1 gene and suggest that they have limited biological activity and that the principal regulator of Csn1s1 expression in pregnancy lies within the Stat5 of the promoter. Alternatively, the two enhancers could be partially redundant and work in a superadditive fashion, so that only when both enhancers are removed, the effect on Csn1s1 expression would be revealed.

Response

We fully agree with the reviewer and added sentences, including wording proposed by the reviewer. "It is possible that the two enhancers could be partially redundant and work in a superadditive fashion, so that only when both enhancers are removed, the effect on Csn1s1 expression would be revealed (see Discussion). Alternatively, the two candidate enhancers could synergize with the STAT5-based promoter element and control Csn1s1 expression during pregnancy and lactation" (lines 257 – 261). We also elaborate on the usage of different enhancers in the Discussion (lines 416 – 419).

Lines 272/273/274: The authors suggest that there might be differences in the promoter and enhancer binding of the transcription factors, since they do not observe a classical gap or nucleosome free region at the promoter. At the resolution shown, this is not possible to judge, alternatively, the promoter region could be more flexible and show a higher heterogeneity in the TF binding. In a population wide assay such as ChIP-seq this is hard to judge without further data.

Response

We fully agree with the possibility of such a scenario and added a sentence including the suggested alternative explanation to the manuscript. "It is further possible that promoter regions are more flexible and present higher heterogeneity in TF binding and different chromatin organization due to the gene transcription or 3D organization of these genes can be envisioned (lines 284 – 287).

REVIEWERS' COMMENTS

Reviewer #2 (Remarks to the Author):

The authors have addressed all concerns, thank you.

Reviewer #3 (Remarks to the Author):

The authors have addressed all my concerns and I would like to congratulate the authors to a great paper that has really evolved during the review process.